# Tsunami evacuation plans for future megathrust earthquakes in Padang, Indonesia considering stochastic earthquake scenarios

Ario Muhammad[1,2], Katsuichiro Goda[1], Nicholas A. Alexander[1], Widjo Kongko[3], and Abdul Muhari[4]

[1]Department of Civil Engineering, University of Bristol, Bristol, BS8 1TR, UK
[2]Department of Civil Engineering, University of Narotama, Surabaya, 60117, Indonesia
[3]Agency for the Assessment & Application of Technology (BPPT), Yogyakarta, 55281, Indonesia
[4]Directorate General for Marine Spatial Management, Ministry of Marine Affairs and Fisheries, Jakarta, 10110, Indonesia

*Correspondence to*: Ario Muhammad (ario.muhammad@bristol.ac.uk)

**Abstract.** This study develops tsunami evacuation plans in Padang, Indonesia, using a stochastic tsunami simulation method. The stochastic results are based on multiple earthquake scenarios for different magnitudes (Mw 8.5, 8.75, and 9.0) that reflect asperity characteristics of the 1797 historical event in the same region. The generation of the earthquake scenarios involves probabilistic models of earthquake source parameters and stochastic synthesis of earthquake slip distributions. In total, 300 source models are generated to produce comprehensive tsunami evacuation plans in Padang. The tsunami hazard assessment results show that Padang may face significant tsunamis causing the maximum tsunami inundation height and depth of 15 m and 10 m, respectively. A comprehensive tsunami evacuation plan, including horizontal evacuation area maps, assessment of temporary shelters considering impact due to ground shaking and tsunami, and integrated horizontal-vertical evacuation time maps, has been developed based on the stochastic tsunami simulation results. The developed evacuation plans highlight that comprehensive mitigation policies can be produced from the stochastic tsunami simulation against the future tsunamigenic events.

## 1 Introduction

Tsunami hazard and risk assessments have become an important issue in tsunami-prone regions especially after the 2004 Aceh-Andaman earthquake ($M_w$ 9.15) and the 2011 Tohoku Japan earthquake ($M_w$ 9.0). Significant risk mitigation efforts have been made in recent years in high-risk countries, such as Japan, U.S.A., Chile, New Zealand, and Indonesia (Schlurmann et al., 2010; Wood et al., 2014; Mueller et al., 2015; Raby et al., 2015). Even though Japan is a well-developed country with comprehensive tsunami defense systems, the 2011 Tohoku tsunami still caused significant damage claiming economic loss of ~365 billion USD and fatalities of ~20,000 people (Kazama and Noda, 2012). Globally, preparedness systems against the earthquake and tsunami hazards need to be improved to reduce the economic and social impact due to future tsunamigenic earthquake events (Scheer et al., 2011; Wood et al., 2012). Among the tsunami-prone countries, Indonesia is in one of the most seismically active zones; there were 34 major tsunamigenic events in the last 20 years (USGS, 2015). Past paleo-geodetic, paleo-tsunami, and geodetic investigations (e.g. Nalbant et al., 2005; Sieh et al., 2008)

indicate that the Mentawai segment of the Sunda subduction zone (see FIGURE 1A) can host large tsunamigenic events ($M_w$ > 8.5) with a recurrence period of about 200 years. The last major tsunamigenic earthquakes in this region were the 1797 and 1833 events (Natawidjaja et al., 2006), while two recent events, $M_w$ 8.4 and $M_w$ 7.9, occurred near Bengkulu on 12 and 13 September 2007 (see FIGURE 1A). A study by Konca et al. (2007) concluded that the recent earthquakes released far smaller amounts of slip in comparison with the accumulated slip since 1833 and hence, the potential of a large tsunamigenic event originated from this source remains high.

Padang is a home of more than 850,000 and is one of the most urbanized cities in western Sumatra. The 2004 Aceh-Andaman tsunami did not significantly affect this region since the source location of the 2004 event was far, i.e. >1,200 km (Natawidjaja et al., 2006; Meltzner et al., 2006; Briggs et al., 2006). However, it is located along the coast of Sumatra Island, directly facing the Mentawai segment of the Sunda subduction zone (see FIGURE 1A). Consequently, potential impact of the future tsunami may be significant in this area. In addition, with the low-lying plain topographic features in Padang, the probability of large inundated areas and large inundation depths is also high (Borrero et al., 2006; Muhari et al., 2010, 2011). In the past, two types of earthquake source scenarios have been mainly considered to develop tsunami risk mitigation plans in Padang: deterministic scenarios (Borrero et al., 2006; Schlurmann et al., 2010; Muhari et al., 2010, 2011) and probabilistic scenarios (McCloskey et al., 2008; Griffin et al., 2016). These two methods have both advantages and disadvantages. For instance, the deterministic approach is more communicable to the authorities for developing post-disaster recovery and mitigation plans (McGuire, 2001). However, implementation of deterministic scenarios may oversimplify the tsunami hazards and risks, leading to imprecise mitigation plans (Mueller et al., 2014; Griffin et al., 2015). On the other hand, the probabilistic scenario approach requires the proper consideration of regional earthquake characteristics, including uncertainties in size of the rupture plane and spatial heterogeneity of earthquake slip. Therefore, extensive and detailed data regarding the regional seismological characteristics are essential to develop the probabilistic scenarios. In the previous investigations, those regional earthquake characteristics have not been taken into account properly. Recently, Muhammad et al. (2016) have evaluated the tsunami potential in Padang by developing the stochastic tsunami simulation method allowing to generate numerous scenarios of stochastic tsunami hazard. However, that work was limited to evaluate the tsunami hazards off-shore and near coast only because of the gross bias of the elevation model in Padang (Griffin et al., 2015) and therefore, was not suitable to carry out rigorous assessment of tsunami mitigation systems. The gross bias of the elevation model in Padang is due to the use of global Digital Elevation Model (DEM; i.e. GDEM2) as the elevation data for the stochastic tsunami simulation. The absolute vertical errors of global DEM (e.g. GDEM2 and SRTM1) are in the range from 5 m to 10 m resulting in inaccurate prediction of the inundation footprints especially in coastal areas (Sanders, 2007; Gallegos, 2009; Lewis et al., 2013; Griffin et al., 2015). Thereby, this work also highlights the effect of DEM on inundation modeling in Padang areas which has not been studied in the previous investigations.

Moreover, an effective tsunami evacuation plan should combine both horizontal evacuation to high grounds and vertical evacuation to designated tsunami-resistant shelters (FEMA P-646, 2012). In the coastal areas where people can afford relatively short evacuation time (less than 30 minutes), the vertical evacuation is highly desirable (Scheer et al., 2012;

Wood et al., 2014). In the previous investigations, the tsunami arrival time in Padang was estimated to be 20-30 minutes (Borrero et al., 2006; McCloskey et al., 2008; Schlurmann et al., 2010; Muhammad et al., 2016) and therefore, effective vertical evacuation plans are needed in this region. In Padang, 23 tsunami evacuation shelters (TES) have been planned and built in the urban areas near the coastal line. However, an extensive assessment of the TES in Padang during the tsunami event was not conducted by the previous studies (Schlurmann et al., 2010; Muhari et al., 2010, 2011; Imamura et al., 2011).

Only horizontal evacuation time maps to safe areas were provided in Schlurmann et al. (2010) excluding seismic and tsunami vulnerability assessments of TES. Subsequently, it is important to assess the TES considering impacts of ground shaking and tsunami in Padang as part of future tsunami mitigation planning. The results of the TES assessment can be further used to develop integrated horizontal-vertical tsunami evacuation maps.

    Building upon the previous studies, this study develops tsunami evacuation plans in Padang based on the stochastic

tsunami simulation method. This approach generates multiple earthquake scenarios by considering the uncertainties of the earthquake source parameters and slip distributions, both of which have major influence on the tsunami hazards. Hence, it is suitable to estimate the tsunami hazard level in Padang. Regional seismological characteristics are taken into account in generating stochastic earthquake scenarios based on the finite-fault models of the past earthquakes in the Sunda subduction zone. In addition, a 5-m high-resolution DEM of Padang (DEM5) developed by the German-Indonesian Tsunami Early

Warning System (GITEWS) project and Indonesian research institutions is adopted as land elevation data for tsunami simulation. Combination of the stochastic tsunami inundation modeling and the high-resolution DEM significantly improves the accuracy of tsunami inundation modeling in Padang. Subsequently, the tsunami mitigation plans in Padang, i.e. tsunami inundation maps, TES assessments considering impacts due to ground shaking and tsunami, and integrated horizontal-vertical evacuation time maps, are produced based on inundation depths estimated from the stochastic tsunami simulations.

The development of such tsunami mitigation systems in Padang will overcome the limitations of the previous works and will contribute to improving the tsunami preparedness against the future catastrophic events.

## 2 Methodology

### 2.1. Tsunami modeling

### 2.1.1 Earthquake scenario selection

The stochastic tsunami simulation method (Muhammad et al., 2016) is adopted to estimate the tsunami hazard level in Padang. To generate earthquake source model stochastically, earthquake scenarios in terms of magnitude and source region need to be set up in advance. An appropriate model of fault rupture zone including geometry of the fault plane and asperity regions is then defined (see FIGURE 2). The geometry is essential to outline the earthquake source zone, whilst a so-called asperity zone determines the areas of concentrated slips within the fault plane. In general, modeling an earthquake rupture

process in terms of earthquake source and asperity zones for the future tsunamigenic earthquakes in the Mentawai-Sunda

region is complicated and has significant uncertainty (Natawidjaja, 2006; Griffin et al., 2016). In this study, the future earthquake source area in the Mentawai segment is defined using the fault rupture areas of the historical subduction earthquakes in the Sunda subduction zone (see Muhammad et al., 2016 for details).

Firstly, a generic fault model covering the entire region of the Mentawai segment with a size of 920 km in length and 250 km in width is constructed (see FIGURE 1B). Numerous megathrust earthquakes can be further generated within the generic fault model. This generic fault plane is consistent with the results of extensive geodetic, paleo-geodetic, paleo-seismic, and numerical studies (Nalbant et al., 2005; Natawidjaja et al., 2006; Chlieh et al., 2008; Sieh et al., 2008; Philibosian et al., 2014). The depth of fault plane ranges from 3 km to 50 km with a regular strike angle (i.e. 325°). The dip angle varies from 8° to 16° depending on the depth, e.g. the dip angle becomes steeper as the depth increases. The properties of the fault plane are consistent with the previous investigations. For instance, the depth is comparable with the source models of the past Mentawai-Sunda subduction earthquakes and the strike/dip angles are in line with the slab models of the Sunda subduction zone developed by the USGS. (Newman et al., 2011; Satake et al., 2013; Philibosian et al., 2014; Yue et al., 2014; Hayes et al., 2009, 2012). In addition, to stochastically generate the earthquake sources, the fault plane is divided into 10 km by 10 km sub-faults (see FIGURE 1B).

Secondly, within the fault plane of the source zone, asperity zones are set up. The asperity zones reflect the regional seismological knowledge of earthquake ruptures. Understanding the rupture process of past seismicity in the Mentawai segment is essential to determine the asperity zones. Based on the past seismicity, the most likely asperity zones for the future tsunamigenic earthquake events in the Mentawai segment are located in the rupture areas of the 1797 and 1833 events since these two events were the last tsunamigenic earthquake events in the Mentawai-Sunda region (Borrerro et al., 2006; Natawidjaja et al., 2006; McCloskey et al., 2008). An area having 300 km long extended from 0.5°S to 3.2°S of the Mentawai segment (see FIGURE 1A) was determined as the asperity zone of the 1797 event based on the geodetic and paleo-geodetic measurements. On the other hand, an area having 320 km long extended from 2.1°S to 5°S (see FIGURE 1A) was inferred as the asperity zone of the 1833 event (Natawidjaja et al., 2006; Philibosian et al., 2014). Note that the 1797 event produced more significant tsunami impacts in Padang than the 1833 event (Borrero et al., 2006; Natawidjaja et al., 2006; McCloskey et al., 2008). Consequently, in this study, the 1797 asperity zone is adopted to generate the future stochastic earthquake source models. In addition, in terms of selected magnitude scenarios, three scenario magnitudes are considered: $M_w$ 8.5, $M_w$ 8.75, and $M_w$ 9.0. The magnitude $M_w$ 8.5 is used as the minimum scenario because the tsunami hazard produced from the magnitude below this level, e.g. $M_w$ 8.25 and $Mw$ 8.0, is relatively small (less than 1 m wave height in the coastal areas; see Muhammad et al., 2016). The maximum magnitude scenario ($M_w$ 9.0) is selected based on geodetic, paleo-geodetic, and paleo-tsunami studies (Zachariasen et al., 1999; Natawidjaja et al., 2006; Sieh et al., 2008); they indicated that the accumulated slip in the Mentawai segment of the Sunda subduction zone may generate the tsunamigenic earthquake with the magnitude range from $M_w$ 8.8 to $M_w$ 9.0.

### 2.1.2 Stochastic tsunami simulation

The stochastic tsunami simulation involves two main processes: (1) stochastic earthquake source model generation and (2) Monte Carlo tsunami simulation (see FIGURE 2). For generating realistic source models stochastically, regional seismological characteristics of Sumatra are analyzed. It is carried out by estimating earthquake source properties including geometry (fault length, $L$ and fault width, $W$), slip statistics (mean slip, $D_a$, maximum slip, $D_m$, and Box-Cox parameter, $\lambda$), and spatial heterogeneity parameters (correlation length along strike direction, $A_x$, correlation length along dip direction, $A_z$, and Hurst number, $H$). Those parameters are calculated using the effective dimension analysis method (Mai and Beroza, 2000), Box-Cox analysis (Goda et al., 2014), and spectral analysis (Mai and Beroza, 2002); see Muhammad et al. (2016) for the details.

Subsequently, the calculated regional earthquake source parameters are compared against the global scaling relationships developed by Goda et al. (2016) to examine the adequacy of the global models to the Mentawai-Sunda region. Muhammad et al. (2016) concluded that the regional earthquake source parameters calculated from the 19 past Sunda earthquakes are in good agreement with these scaling relationships; subsequently, the global models are adopted in this study. A set of 100 source models is then generated for each magnitude. Therefore, the total number of the stochastic earthquake slip models used in this study is 300.

For a given stochastic source model, the initial deformation of seabed is calculated by considering both horizontal and vertical displacements of the seafloor using Okada (1985) and Tanioka and Satake (1996). Tsunami modeling is then carried out by solving non-linear shallow water equations with run up (Equations 1 to 3). A finite-difference method implementing a staggered leap-frog scheme is adopted to solve the governing equations (Goto et al., 1997). In addition, in Goto et al.'s code the moving boundary approach developed by Iwasaki and Mano (1979) is used for inundation modeling. This method has been successfully used to run the tsunami simulation in several regions, including Padang, Indonesia, Mexico, and Japan (Muhari et al., 2010, 2011; Goda et al., 2014; Mori et al., 2017).

$$\frac{\partial \eta}{\partial t} + \frac{\partial M}{\partial x} + \frac{\partial N}{\partial y} = 0 \; , \tag{1}$$

$$\frac{\partial M}{\partial t} + \frac{\partial}{\partial x}\left(\frac{M^2}{D}\right) + \frac{\partial}{\partial y}\left(\frac{MN}{D}\right) + gD\frac{\partial \eta}{\partial x} + \frac{gn^2}{D^{7/3}}M\sqrt{M^2+N^2} = 0 \; , \tag{2}$$

$$\frac{\partial N}{\partial t} + \frac{\partial}{\partial x}\left(\frac{MN}{D}\right) + \frac{\partial}{\partial y}\left(\frac{N^2}{D}\right) + gD\frac{\partial \eta}{\partial y} + \frac{gn^2}{D^{7/3}}N\sqrt{M^2+N^2} = 0 \; , \tag{3}$$

where $D = h + \eta$ representing the total water depth, in which $h$ and $\eta$ are the water depth and the tsunami height above the reference sea level, respectively, $g$ is the gravitational acceleration, and $n$ is the Manning's roughness coefficient. The discharge fluxes (i.e. $M$ and $N$) are obtained from the integration of velocity in x ($u$) and y ($v$) directions over the water depth (Equations 4 and 5).

$$M = \int_{-h}^{\eta} u \, dz = u(h+\eta) = uD \; , \tag{4}$$

$$N = \int_{-h}^{\eta} v \, dz = v(h + \eta) = vD , \tag{5}$$

Moreover, DEM and bathymetry datasets are developed to run the stochastic tsunami simulation. In this study, the DEM5 is adopted as land elevation data (see Section 2.1.3 for the details), whilst for bathymetry, the GEBCO2014 dataset (http://www.gebco.net/data_and_products/gridded_bathymetry_data/) and a 3-m Padang bathymetry dataset are combined. Four nested grids, i.e. 1350 m, 450 m, 150 m, and 50 m, being produced from linear interpolation of these datasets, are used to run the tsunami simulation. A roughness coefficient of 0.025 $m^{-1/3}$s for water and 0.06 $m^{-1/3}$s for land are adopted to model the surface roughness effects on tsunami flows (Griffin et al., 2015). By assuming instantaneous fault rupture, the duration and time step of tsunami simulation are then defined as 2 hours and 0.5 s, respectively. It satisfies the Courant-Friedrichs-Lewys (C.F.L.) criterion for the bathymetry and elevation data for the Mentawai region. The Monte Carlo tsunami simulation is finally performed using different stochastic earthquake source scenarios.

### 2.1.3 Effect of Digital Elevation Model on tsunami inundation modeling

An accurate DEM is essential for tsunami simulation and is particularly critical in calculating tsunami inundation depths. Recently, several international organizations and consortia have produced global DEM datasets, including GTOPO30, SRTM30, SRTM3v2, SRTM3v4, SRTM1, and GDEM2. Currently, the SRTM1 (https://lta.cr.usgs.gov/SRTM1Arc) and GDEM2 (https://asterweb.jpl.nasa.gov/gdem.asp) with the resolution of 1 arc-sec (~30 m) are the best available global DEM datasets and widely used for land elevation data in tsunami simulation (Satake et al., 2013; Yue et al., 2014). In general, the range of numerical processes employed in the compilation of DEMs can have a major influence on the magnitude of errors. Both epistemic and aleatory noise can be present. For instance, the average vertical errors of SRTM1 and GDEM2 are in the range of 10 m (Tachikawa et al., 2011; Satge et al., 2015). Because tsunami evacuation plans are mainly developed based on the inundation results, the effect of DEM on tsunami inundation modeling is assessed before presenting the main tsunami simulation results in Section 3. In this study, the baseline DEM is the local DEM5. The DEM5 dataset was developed from GPS measurements and high-resolution satellite imagery to produce the DEM profile of Padang, particularly near the coastal line (see FIGURE 4A). A vertical error of 0.2 m was found in this dataset. Hence, the DEM5 is reliable to represent the land elevation of Padang (Taubenbock et al., 2009; Schlurmann et al., 2010).

In this section, two global DEM datasets, i.e. SRTM1 and GDEM2, are adopted to study the effect of DEMs on tsunami inundation modeling in the Padang areas (see FIGURE 4B and FIGURE 4C). The elevation differences between the DEM5 and the two global datasets (i.e. SRTM1 and GDEM2) are firstly assessed and discussed. Then, the differences on tsunami inundation results from these three datasets are further presented. Note that the bathymetry dataset (GEBCO2014) is common for all three cases. The considered source model is one realization from the 100 stochastic earthquake sources in the Mentawai-Sunda zone for the $M_w$ 9.0 scenario. This is shown in FIGURE 3B; the maximum slip of the chosen model reaches 25 m with the fault size of 450 km in length and 250 km in width.

The elevation differences of the SRTM1 and the GDEM2 datasets with respect to the reference DEM5 data are presented in FIGURE 5A and FIGURE 5B, respectively. The elevation differences are calculated by taking the elevation differences between the global DEM datasets (i.e. SRTM1 and GDEM2) and the reference data. Several statistics including the minimum, maximum, mean difference, absolute mean difference, and root mean square error (RMSE) values, are calculated for each global DEM dataset and presented in Table 1. The difference with the reference data in terms of maximum elevation value for SRTM1 is 7 m. This is smaller than 12 m for GDEM2. The other three statistical scores, i.e. the mean difference, the absolute mean difference, and the RMSE for SRTM1 are also smaller than GDEM2. For instance, the RMSE of SRTM1 is 4.27 m, which is smaller than 7.46 m for GDEM2. In general, SRTM1 performs better than GDEM2. However, the RMSE scores of 4.27 m and 7.46 m may lead to inaccurate modeling of inundation in land especially in low-lying areas where the elevation is below 10 m.

Subsequently, to evaluate the effect of DEMs on inundation modeling, FIGURE 7 illustrates inundation maps in Padang based on three different DEMs, i.e. DEM5 (FIGURE 7A), SRTM1 (FIGURE 7B), and GDEM2 (FIGURE 7C). Total inundation areas are also presented in FIGURE 7. The inundation maps indicate that the global DEM datasets (i.e. SRTM1 and GDEM2) underestimate the inundation areas significantly. The total inundation area of SRTM1 is less than a half of DEM5 (i.e. 7.32 km$^2$ in comparison to 16.04 km$^2$ for DEM5). On the other hand, GDEM2 performs worse than SRTM1 with only 0.74 km$^2$ of total inundation area. The inundation difference for SRTM1 is mainly because of relatively high elevation differences (~5 m) between SRTM1 and DEM5 near the coastal region (see FIGURE 5A). Moreover, for GDEM2, the significant differences are mainly due to the overlapping land areas in the seaside area of the GDEM2 dataset which prevent the tsunami flow to go further in land (see black rectangle in FIGURE 4C). In addition, significant differences of inundation areas found in the GDEM2 dataset are also due to high RMSE (7.42 m) and elevation differences (~8 m) in low-lying areas near the coastal line of Padang. Consequently, it is highly recommended to use an accurate local elevation dataset merged for tsunami inundation modeling and hence, the DEM5 dataset is adopted in this study as land data.

## 2.2 Vulnerability assessment of tsunami evacuation shelters against ground shaking and tsunami

Prior to developing tsunami evacuation plans, structural vulnerability of TES in Padang needs to be evaluated considering both ground shaking and tsunami hazards. Suitable seismic and tsunami fragility models are used to assess the vulnerability of the TES by determining the probability that a building sustain a certain damage state for a given earthquake scenario. Since the ground shaking affects the TES before the tsunami, the seismic vulnerability assessment of TES is carried out first, followed by the tsunami vulnerability assessment. In this study, the combined effects of earthquake shaking and tsunami are not taken into account, because such multi-hazard fragility models are not available for TES in Padang. Detailed procedures for the earthquake-tsunami vulnerability assessments of TES are presented in the following sections.

### 2.2.1 Earthquake simulation

Seismic intensity measures for a given earthquake scenario at a TES can be effectively estimated using ground motion prediction equations (e.g. www.gmpe.org.uk). Because the capacity spectrum method (CSM) together with seismic fragility models is used for seismic vulnerability assessment (Section 2.2.2), elastic spectral accelerations (Sa) over a range of vibration periods need to be simulated for various earthquake scenarios from the Mentawai-Sunda region. In this study, among existing ground motion models, a relationship by Abrahamson et al. (2016) is adopted because it is developed and

validated based on extensive global subduction ground motion databases (including ground motion data from the 2010 Maule Chile and 2011 Tohoku Japan earthquakes) and is applicable to mega-thrust interface subduction earthquakes in Sumatra, Indonesia.

The formula to model ground motion intensities due to interface subduction earthquakes is given by:

$$\ln(S_a) = \theta_1 + \theta_4 \cdot \Delta C_1 + [\theta_2 + \theta_3 \cdot (M_w - 7.8)] \cdot \ln\{R + C_4 \cdot \exp[\theta_9 \cdot (M_w - 6)]\} + \theta_6 \cdot R + f_{MAG}(M_w) + f_{FABA}(R) +$$

$$f_{SITE}(PGA_{1000}, V_{s30}) + \sigma \cdot \varepsilon \tag{6}$$

where ln is the natural logarithm, $M_w$ is the moment magnitude, R (km) is the closest distance to the rupture plane, $V_{S30}$ (m/s) is the average shear wave velocity in the uppermost 30 m of surface soil, $PGA_{1000}$ is the median peak ground acceleration (PGA) value corresponding to $V_{S30} = 1000$ m/s, $\sigma$ is the total standard deviation, and $\varepsilon$ is the Gaussian error term represented by zero mean and unit standard deviation. The magnitude function is given by:

$$f_{MAG}(M) = \begin{cases} \theta_4 \cdot [M_w - (7.8 + \Delta C_1)] + \theta_{13} \cdot (10 - M_w)^2 & for\ M_w \leq 7.8 + \Delta C_1 \\ \theta_5 \cdot [M_w - (7.8 + \Delta C_1)] + \theta_{13} \cdot (10 - M_w)^2 & for\ M_w \leq 7.8 + \Delta C_1 \end{cases} \tag{7}$$

where $\Delta C_1$ captures the epistemic uncertainty related to the break in magnitude scaling. $f_{FABA}(R)$ is the forearc/backarc term which is equal to 0 for forearc or unknown site and one for backarc. Because Padang is in the forearc region of the Sumtara subduction zone, $f_{FABA}(R)$ is set to 0. Moreover, the site response scaling term is given by:

$$f_{SITE} = \theta_{12} \cdot \ln\left(\frac{\min(V_{S30}, 1000)}{V_{lin}}\right) - b \cdot \ln(PGA_{1000} + c) + b \cdot \ln\left[PGA_{1000} + c \cdot \left(\frac{\min(V_{S30}, 1000)}{V_{lin}}\right)^n\right] \quad for\ V_{S30}$$

$$< V_{lin}$$

$$f_{SITE} = \theta_{12} \cdot \ln\left(\frac{\min(V_{S30}, 1000)}{V_{lin}}\right) + b \cdot n \cdot \ln\left(\frac{\min(V_{S30}, 1000)}{V_{lin}}\right) \quad for\ V_{S30} \geq V_{lin} \tag{8}$$

All model coefficients included in Eqs. 6-8 can be found in Abrahamson et al. (2016).

To simulate ground motion intensities using the Abrahamson et al. model, three parameters are needed as inputs: magnitude, rupture distance, and average shear wave velocity for the considered site. For the TES assessment, only the worst

magnitude earthquake scenario is considered ($M_w$ 9.0), whilst the rupture distance is determined based on the closest distance between the location of interest and the rupture fault plane. Since the locations of TES are relatively close one another (the maximum distance among the TES is less than 3 km) and this is significantly smaller than the distance between Padang and the earthquake source region, seismic vulnerability assessment is conducted for a single representative site in Padang. Subsequently, the shortest and longest source-to-site distances are calculated as 55 km and ~100 km, respectively,

among the 100 earthquake scenarios. Moreover, $V_{S30}$ values in coastal areas of Padang range from 200 m/s to 400 m/s (Putra et al., 2014) and hence, $V_{S30}$ of 300 m/s is used in this study. Finally, to include the uncertainty of the prediction equation for multiple spectral acceleration ordinates, a multivariate lognormal distribution is adopted. The median values of spectral accelerations at different vibration periods (at a site of interest) are evaluated using the ground motion model with the three parameters, whereas their covariance matrix of the prediction error terms (i.e. $\sigma\varepsilon$) are based on inter-period correlation of ground motion parameters $\rho(T_1, T_2)$ (see Baker and Cornell, 2006). The correlation coefficient matrix has diagonal elements equal to 1 and off-diagonal elements equal to the correlation coefficient $\rho(T_1, T_2)$, and is calculated based on the following equation (Goda and Atkinson, 2009):

$$\rho(T_1, T_2) = \frac{1}{3}\left(1 - cos\left\{\frac{\pi}{2} - \left[\theta_1 + \theta_2 I_{T_{min}<0.25} \times \left(\frac{T_{min}}{T_{max}}\right)^{\theta_3} log_{10}\left(\frac{T_{min}}{0.25}\right)\right] log_{10}\left(\frac{T_{max}}{T_{min}}\right)\right\}\right) + \frac{1}{3}\left\{1 + cos\left[-1.5 log_{10}\left(\frac{T_{max}}{T_{min}}\right)\right]\right\} \tag{9}$$

where $\theta_1$, $\theta_2$, and $\theta_3$ are the model parameters ($\theta_1 = 1.374$, $\theta_2 = 5.586$, and $\theta_3 = 0.728$), $T_{max}$ and $T_{min}$ are the maximum and the minimum value of $T_1$ and $T_2$, respectively, and $I_{T_{min}<0.25}$ is the indicator function that equals one if $T_{min} < 0.25$ sec and equals zero otherwise. Equation 9 was developed based on subduction earthquake records from Japan; thus, it is considered to be applicable to subduction earthquakes in Sumatra.

### 2.2.2 Seismic vulnerability assessment

The seismic vulnerability of TES can be assessed using seismic fragility models. The fragility models relate the probability that a building's damage state exceeds a particular threshold for a ground motion parameter, such as PGA and Sa (Rossetto and Elnashai, 2003). In general, the seismic damage states (DS) can be categorized as slight (DS1), moderate (DS2), extensive (DS3), and complete (DS4; see **Table 3**). Typically, for DS1, fine cracks in plaster partitions/infills to hairline cracking in beams and columns near joints (<1 mm) are found. For DS2, cracking occurs in most of the beams and columns along with larger flexural cracks and concrete spalling. On the other hand, for DS3, some elements of the building may reach their ultimate capacity revealed as large flexural cracking, concrete spalling, and re-bar buckling (Rosetto and Elnashai, 2003; see **Table 3**). The damage states for tsunami differ from those for ground shaking. Subsequently, in this study, the seismic fragility curves developed for and implemented in HAZUS (2003) is adopted to assess the vulnerability of TES in Padang because of the following reasons:

1. The TES are designed and constructed according to the new Indonesian Earthquake Resistance Building Code (SNI-1726: 2012) adopting the U.S. seismic design documents, i.e. FEMA P750 (2009), regarding seismic design provisions for new building and other structures, and ASCE/SEI 7-10 for the minimum design load criterion (SNI-1726: 2012; Kurniawan et al., 2014; Wijayanti et al., 2015; Aulia, 2016; Sengara et al., 2016).

2. HAZUS is a well-established earthquake loss estimation framework and has been implemented in several earthquake-prone countries for seismic risk assessment purposes, e.g., Haiti, Puerto Rico, France, Romania, Austria, and Indonesia (Kulmesh, 2010; Peterson and Small, 2012; Wijayanti et al., 2015; Sengara et al., 2016).

In HAZUS, building performance under seismic actions is evaluated based on the CSM. The CSM compares the structural capacity in terms of capacity curve with the seismic demand on a structure that account for post-yielding inelastic behavior of the structural system. The non-linear inelastic behavior of the structural system is taken into account by applying the effective reduction factor to the elastic response spectrum of the considered earthquake scenario (Freeman, 2004; Kim et al., 2005; Monteiro et al., 2014). The performance point, which is the expected seismic response of a structural system for a given earthquake scenario, can be obtained graphically as an intersection point of the capacity and demand curves. **FIGURE 6**A illustrates the procedure for developing an inelastic response (demand) spectrum from the elastic response (input) spectrum in HAZUS. First, the acceleration response spectrum is generated from the earthquake simulation (see Section 2.2.1), and is further converted into the acceleration-displacement response spectrum (ADRS). In the CSM, the ADRS is defined as the elastic response spectrum (ERS). Second, the inelastic demand spectrum is calculated by dividing the ERS by the reduction factors (i.e. $R_A$ at periods of constant acceleration and $R_V$ at periods of constant velocity). Note that the reduction factors in HAZUS are equal to the reciprocal of $SR_A$ and $SR_V$ in ATC-40 (ATC, 1996). For essential and average buildings (type B), the $SR_A$ and $SR_V$ should be less than 2.27 and 1.79, respectively (ATC, 1996). On the other hand, the TES may be classified as type B based on the ATC-40 system and hence, $R_A$ and $R_V$ should be less than 2.27 and 1.79, respectively. In this study, both $R_A$ and $R_V$ are set to 1.5 (Lin and Chang, 2003; Casarotti et al., 2009; Monteiro et al., 2014). Third, the capacity curve taken from HAZUS is overlaid to compare with the inelastic response spectrum (see blue line in **FIGURE 6**A). The capacity curve in HAZUS is defined based on two parameters, e.g. yield and ultimate strengths characterizing the nonlinear (pushover) behavior. The building-type classifications in HAZUS are based on the building material (e.g. wood, reinforced concrete and steel) and height. Following the HAZUS classification, the TES in Padang is categorized as reinforced concrete moment resistant frames (RC-MRF) with different building heights. TES numbers 13 and 16 are high-rise RC-MRF (C1H), whereas the rest of TES are mid-rise RC-MRF (C1M). Moreover, in HAZUS, four seismic design codes classification including Pre-Code, Low-Code, Moderate-Code, and High-Code are defined corresponding to the seismic zone. In terms of seismic design code classification, High-Code is applicable to TES in Padang, because Padang is located in the high seismic zone and TES has been designed and constructed to higher standards/quality than other normal buildings (Kurniawan et al., 2014; Aulia, 2016). In the following, the seismic vulnerability assessment of TES is carried out by focusing upon C1M because the C1H type is typically stronger than C1M in terms of capacity curve (i.e. for the same shaking intensity, CH1 buildings are expected to perform better than CM1).

Finally, seismic fragility curves implemented in HAZUS are used to define the damage functions of the building; typically, the fragility functions are defined using the lognormal distribution. Four damage states, i.e. slight, moderate,

extensive, and complete, are defined in HAZUS (see **Table 3** and **FIGURE 6**B for the descriptions). Subsequently, to determine whether a TES can be used for post-earthquake tsunami evacuation purposes (not for shelters), the building is categorized into safe and unsafe by referring to existing tagging criteria (FEMA 356, 2000; HAZUS, 2003; Bazzurro et al., 2006) including (see **FIGURE 6**B):

- Green tag: the building may have experienced onset damage but is safe for immediate occupancy. The none-to-slight damage state is applicable.
- Yellow tag: re-occupancy of the building is restricted and limited access only is allowed. Moderate-to-extensive damage state corresponds to this case.
- Red tag: the building is unsafe and no access is granted, and will be in complete damage or collapse state.

Based on the above tagging criteria, the tsunami evacuation building may be judged as unsafe for evacuation if the probability of extensive and complete damage states is over 50%. This assumption gives a 50-50 chance that the building may experience above or below extensive damage (Bazzurro et al., 2006). Moreover, the 50% probability of extensive or severer damage state is typically identified as the threshold value of a yellow tag in HAZUS that is adopted in this study (see **FIGURE 6**B) and hence, may be regarded as the limit state to define the accessibility of buildings for emergency evacuation
during the tsunami inundation.

### 2.2.3 Tsunami vulnerability assessment and TES capacity estimation

Once the TES is judged to be accessible for tsunami evacuation through the seismic vulnerability assessment, a tsunami vulnerability assessment is then carried out. The tsunami fragility models evaluate the probability of experiencing certain damage states for a building due to tsunami. The tsunami damage criteria for buildings are defined by the Japanese Ministry
of Land Infrastructure Tourism and Transport (2013), which are shown in Table 4.

In current literature, the majority of tsunami fragility models have been developed using post-tsunami survey data, remote sensing data, and numerical modeling (e.g. Dias et al., 2009; Koshimura et al., 2009; Suppasri et al., 2011). In this study, the model by Suppasri et al. (2011) is adopted for the following reasons. It was developed through extensive remote sensing and tsunami survey data (i.e. ~5,000 points) in Banda Aceh and Thailand for the 2004 Aceh-Andaman tsunami, and
335 is the most recent model among existing tsunami fragility models that are applicable to Sumatra, Indonesia. These features are important because current situations of tsunami mitigation measures in Padang resemble those in Banda Aceh and Thailand more closely than situations in other regions. The Suppasri et al. model considers three damage states for tsunami damage (see **FIGURE 6**C) and consists of three fragility curves for reinforced concrete building for slight (DST1), moderate (DST2), and major/severe damage state (DST3). Using the calculated probability exceedance of each damage state, the TES
is considered to be unsafe if the exceedance probability of severe damage is above 50% (the major tsunami damage is assumed to be similar to the extensive damage in seismic damage state criteria).

 the TES capacity for accommodation is calculated. This is evaluated by calculating the TES capacity (TESC):

$$TESC = (EAF \times NF)/SpP, \tag{10}$$

where $EAF$ is the existing evacuation area at each floor and $NF$ is the total number of floors excluding inundated floor (Budiarjo, 2006; Widyaningrum, 2009; Dewi, 2012). Several assumptions are made to calculate the TES building capacity, which are the following:

1. Floors of the TES buildings are categorized into two floor types, i.e. unsafe floor and evacuation floor. The unsafe floor is the floor that is inundated by the tsunami, whilst the evacuation floor is the floor for evacuation areas. Moreover, the inundated floor is considered to be unsafe for evacuation and hence, is excluded from building capacity estimation during the tsunamigenic event.

2. Space needed per person ($SpP$) at the evacuation areas in each TES building is 1 m$^2$ determined based on 0.8 m$^2$ for stay and 0.2 m$^2$ for circulation (BAPPENAS, 2005; Budiarjo, 2006; Widyaningrum, 2009; Dewi, 2012). This value is similar to that recommended by (FEMA P-646, 2012), i.e. 0.93 m$^2$ per person.

3. Existing evacuation area in each floor of the TES building is assumed to be equal for all floors because only total evacuation area data for the whole building are available.

## 2.2.4 Evacuation time maps

Next, horizontal and vertical tsunami evacuation time maps are developed based on the total evacuation time (TET), which is calculated by summing initial reaction time (IRT) and evacuation time (ET).

$$TET = IRT + ET, \tag{11}$$

$$IRT = DT + NT + RT, \tag{12}$$

The initial reaction time is the actual response time for the community to start the evacuation, whilst the evacuation time is the time needed for the community to evacuate to the safe areas. In principle, three components are considered to calculate the initial reaction time (IRT) during the tsunamigenic event including institutional decision time (DT), institutional notification time (NT), and reaction time of the community (RT), as presented in Equation 12). The institutional times (DT and NT) are determined by the related government institution which has the authority to issue hazard warning (Charnkol and Tanaboriboon, 2006; Post et al., 2009). In Indonesia, the official institution to release tsunami warning is the Indonesia Tsunami Early Warning System of Indonesian Agency for Meteorology, Climatology and Geophysics (INA-TEWS of BMKG). The INA-TEWS normally needs 5 minutes to issue tsunami warning (Widyaningrum, 2009; Dewi, 2012). In addition, the institutional notification time is assumed to be 3 minutes, whilst the reaction time of the community is 7-10 minutes (Charnkol and Tanaboriboon, 2006; Post et al., 2009). In this study, the initial reaction time used is 15 minutes by adopting the community reaction time of 7 minutes. Moreover, a suitable range of the travel speed is found to be 0.91 m/s to

3.83 m/s, depending on the traveling method (by foot or vehicle) and the evacuees' age (Wood et al., 2012; Fraser at al.,

2014). In this study, the evacuation time is calculated based on the slowest travel speed (0.91 m/s) to capture the worst

scenario. The evacuation time is estimated crudely by excluding roads leading to the safe places and other essential

parameters (e.g. population density and age classification). Although the approximate method is not able to capture the

realistic situation of evacuation accurately, it is considered to be useful for emergency managers (e.g. regional disaster

management stakeholders) to develop a city-wide tsunami evacuation plan.

Finally, the integrated evacuation time maps are developed by combining horizontal and vertical evacuation time

maps. The integrated evacuation time maps are calculated by taking a minimum evacuation time between evacuations

horizontally and vertically. These maps are essential for the rescue teams to consider both evacuation options and

subsequently may reduce the casualties during the tsunami event.

## 3 Results

The main results that are discussed in this section are focused on: tsunami hazard level in Padang produced from all

earthquake scenarios $M_w$ 8.5, 8.75, and 9.0 (Section 3.1), vulnerability assessment of TES considering impacts of seismic

and tsunami in Padang using the $M_w$ 9.0 scenario (Section 3.2), and horizontal, vertical, and integrated evacuation time maps

during the tsunamigenic event using the $M_w$ 9.0 scenario (Section 3.3).

### 3.1 Tsunami hazard level in Padang

The tsunami hazard level in Padang is investigated by assessing the tsunami height and depth produced from the stochastic

tsunami simulations for three magnitude scenarios. The height presented in this study is the height of water flow above the

mean sea level, whilst the depth refers to the water flow height above the ground. Firstly, the maximum inundation depth

maps for all scenarios along with the maximum inundation depth maps for the areas above 1 m depth are presented and

discussed. Secondly, the inundation heights along the coastal line and three main rivers in Padang, i.e. (1) Kuranji river, (2)

Banda Bakali river, and (3) Arau river, are discussed (see FIGURE 3A). The inundation footprints along the rivers are

concerned because the tsunami flow may penetrate far inland through the rivers, as observed in the 2011 Tohoku event (Mori

et al., 2011; Tanaka et al., 2014).

The maximum inundation depth maps in Padang are presented in FIGURE 8, whilst the total inundation areas for the

depth above 1 m are shown in FIGURE 9. Three maps in each magnitude scenario are for the 10[th] percentile (left panel), the

50[th] percentile (central panel), and the 90[th] percentile (right panel). FIGURE 8A shows that the tsunami impacts for the case of

$M_w$ 8.5 are insignificant in Padang. The total inundated areas exhibited in the 90[th] percentile map of the $M_w$ 8.5 scenario are

relatively minor (only 3.12 km$^2$). Those inundated areas are concentrated near the coastal line of Padang with the total

inundation areas above 1 m depth is only 0.94 km$^2$ (see the right panel of FIGURE 9A). These results are much smaller than

the maximum tsunami inundation areas for the same magnitude scenario ($M_w$ 8.5) produced by the GITEWS (Goseberg and

Schulrmann, 2009; Taubenbock et al., 2009; Schlurmann et al., 2010). This is due to the use of deterministic source
scenarios in which large earthquake slips are placed very close to Padang (less than 20 km). Consequently, the considered
earthquake scenarios used in the GITEWS project may over-predict the tsunami inundation areas. In addition, the inundation
areas increase significantly from the $M_w$ 8.5 scenario to the $M_w$ 8.75 scenario as shown in the 90th percentile maps. The
maximum total inundation areas for the $M_w$ 8.75 case are about four times larger than the $M_w$ 8.5 scenario (increasing from
3.12 km$^2$ to 11.59 km$^2$). The inundation areas above 1 m depth for the 90th percentile of the $M_w$ 8.75 case also increase
drastically to 8.52 km$^2$ as presented in the right panel of FIGURE 9B.

Moreover, the tsunami effects are found to be much more significant for the $M_w$ 9.0 scenario. The total inundation
areas above 1 m depth reach 16.55 km$^2$ at the 90th percentile. The evacuation from the inundated areas might be very difficult
in such a situation. On the other hand, in general, the maximum tsunami-affected areas produced from the stochastic tsunami
simulations are larger than the results from the GITEWS project which are used to build the current evacuation plan in
Padang (Taubenbock et al., 2009; Schlurmann et al., 2010). The existing tsunami evacuation plan may oversimplify the
future tsunamigenic event and therefore, the improvements of these maps are highly desirable to capture the worst scenarios
that may occur in the future.

To capture the variability of inundation extent in Padang, FIGURE 10 to FIGURE 13 show the inundation height
profiles along the coastal line and three rivers in Padang. The length of the Kuranji, Banda Bakali, and Arau rivers from their
river-mouths are 1.4 km, 1.7 km, and 2.45 km, respectively. In general, high variability of inundation heights is found along
the coastal line of Padang (FIGURE 10). The 10th rank from the three magnitude scenarios show that the tsunami wave
heights along the coastal line range from 0 m (for the $M_w$ 8.5 scenario) to 3 m (for the $M_w$ 9.0 scenario). By contrast, for the
90th rank of the $M_w$ 9.0 scenario, the maximum tsunami height reaches 10 m. Moreover, the inundation heights along the
rivers (see FIGURE 11, FIGURE 12, and FIGURE 13) also show high variability of the wave height, ranging between 0 m and
10 m. The 10th and 50th percentiles of inundation heights for the $M_w$ 8.5 and $M_w$ 8.75 scenarios tend to decrease to zero as the
locations go further inland. However, the tsunami inundation heights remain almost constant along the rivers for the $M_w$ 9.0
case, as presented for the 50th and 90th ranks. This highlights that the tsunami waves can run up the rivers by more than 2 km
from the coastal line and hence, people who live along the rivers may be more affected. In addition, the inundation height
profiles along the coastal line and the rivers in Padang show that the stochastic tsunami simulation method can capture the
uncertainty of the inundation extent. Therefore, the implementation of the stochastic tsunami simulation method for
predicting the future events is highly desirable for preparing more effective and robust mitigation plans.

## 3.2 Vulnerability assessment of tsunami evacuation shelters

Currently, in east and north of Padang, 23 TES have been set up/designated by the National Agency for Disaster
Management (BNPB) of Padang (see Table 2). However, their adequacy as emergency evacuation building was evaluated
under the design scenarios only. Consequently, re-assessment of TES in Padang is highly desirable by taking into account
uncertainties associated with the tsunami hazards. This will provide residents and emergency/rescue teams with valuable

information regarding the current tsunami risk exposure in Padang. In this section, the tsunami inundation depth variability at each TES building is firstly discussed. The vulnerability of TES is then assessed and discussed by considering the worst seismic and tsunami scenarios ($M_w$ 9.0).

First, the variability of tsunami inundation depth at each TES is shown in FIGURE 14. The tsunami impacts to all TES are insignificant in the case of $M_w$ 8.5 for all percentiles. The tsunami depths at the TES locations from 100 tsunami simulations are zero for the majority of the cases. The tsunami impacts start to increase for the $M_w$ 8.75 scenario. In this scenario, the variability of inundation depth at several TES (e.g. shelter numbers 16, 17, 20, and 23) ranges from 0 m to more than 5 m. However, the 90[th] percentile values of tsunami depth for all TES are still below 5 m. On the other hand, the

significant impacts are shown from the variability of tsunami inundation depth for the $M_w$ 9.0 scenario. Three out of the 23 TES, i.e. shelter numbers 1, 15, and 22, with the building height of 10 m may be significantly affected by the tsunami. The depth variability in those three buildings ranges from 0 m to nearly 10 m (see FIGURE 14C). Additional information regarding the depth variability at the TES locations is also presented in FIGURE 15 developed from the $M_w$ 9.0 scenario. Eleven from the 23 TES locations (i.e. shelter numbers 1, 2, 3, 4, 10, 13, 15, 16, 17, 20, and 23) are chosen to illustrate the

depth variability and tsunami arrival time at the TES sites. These stations are located close to the coastal line and the rivers (see FIGURE 3) and hence, are majorly affected by the tsunami. FIGURE 15 shows that several stations (e.g. shelter numbers 16, 17, and 20) are inundated by a maximum depth of nearly 10 m with the arrival times of about 15 minutes based on the 90[th] rank of the $M_w$ 9.0 scenario.

Second, the seismic vulnerability assessment results are carried out for the $M_w$ 9.0 scenario. To illustrate the seismic

vulnerability assessment of TES using HAZUS framework, the median elastic response spectrum for the worst cases of the possible 100 earthquake scenarios (i.e. $M_w$ 9.0 and R = 55 km) is shown in **FIGURE 16**A. Note that this response spectrum does not include the inherent uncertainty associated with the earthquake ground motion simulation. The ADRS for this case is further calculated and shown as a blue line in **FIGURE 16**B. Using the ADRS, the CSM is implemented to determine the performance (demand) point (**FIGURE 16**B). After applying the reduction factors to obtain an inelastic seismic demand

spectrum (green line in **FIGURE 16**B), the performance point is estimated to be about 3 inches (7.6 cm) and then used to calculate the probability exceedance of damage states for a TES. **FIGURE 16**C shows that the sum of probabilities for extensive and complete damage states is ~7% and hence, the TES is considered to be safe for the median response spectra of the worst case.

The assessment that is illustrated in **FIGURE 16A-16C** ignores the inherent uncertainty of input ground motions.

To account for this uncertainty, ground motion simulation results for 100 tsunamigenic earthquake scenarios are presented in **FIGURE 16D-16E** by considering the prediction error terms of the ground motion model together with inter-period correlations. The spectral acceleration profiles show a range of ground shaking that is expected to occur in Padang due to the 100 tsunamigenic earthquakes generated from the Mentawai segment of the Sunda subduction zone. The range of Sa in Padang is between 0.2 g to 1.1 g for the period below 1 s (**FIGURE 16**D). Moreover, the PGA values (**FIGURE 16**E) is at

the interval of 0.3 g to 0.9 g with the median of about 0.5 g. Using the simulated response spectra from those 100 earthquake

scenarios, the TES vulnerability is assessed. **FIGURE 16F** presents the three kinds of exceedance probability of damage states; blue dots correspond to extensive damage state, black dots correspond to complete damage state, and red dots represent the sum of these two probabilities. A 50% probability line is drawn to indicate the threshold of safe building that is considered in this study. **FIGURE 16**F indicates that the TES may be operational for evacuation because ~95% from the total of 100 earthquake simulations produce less than 50% exceedance probability of the combined extensive-complete damage states. Moreover, most of the cases result in less than 25% probability of exceedance above the extensive damage state. Subsequently, the TES may be considered to be safe for evacuation after the ground shaking and hence, the tsunami vulnerability assessment can be carried out.

Fourth, the tsunami vulnerability assessment is performed. Using the maximum inundation depths at all 23 TES from the 100 earthquake scenarios of the $M_w$ 9.0, the probability of exceeding the severe damage state (DST3) for each TES is calculated. When the chance of severe tsunami damage exceeds 50%, the TES is considered to be not usable as tsunami evaluation building. The probabilities of severe damage for the shelter numbers 16 and 17 are relatively large, i.e. 30% and 36% of the 100 events, and hence, these two shelters may be considered to be unsafe for the evacuation (see Table 5). Moreover, the probabilities of severe damage for the other shelters are relatively small (less than 25%). Therefore, except for the shelter numbers 16 and 17, the rest of the shelters are considered to be operational for evacuation.

Subsequently, the estimation of TES building capacity is evaluated. This may capture another point of view regarding the adequacy of existing TES for evacuation. Table 6, presents the estimation of TES building capacity during the tsunami event considering the 10th percentile, the 50th percentile, and the 90th percentile, respectively. In terms of capacity, except for shelters numbers 16 and 17, all TES buildings can be used for vertical evacuation during the 10th rank event. However, for the 50th percentile case, the shelter number 1 (sport center of UNP) may not be operational, whilst for the 90th percentile case, shelter numbers 1 and 15 (Elementary school of 24 Padang) are unable to accommodate evacuees since all floors will be inundated. Note that, for the shelter number 1, there is only one floor since most of the building areas are used for the sport arena. In terms of capacity, for the 50th and 90th rank cases, the possible maximum capacity to be accommodated at all TES buildings are only about 64,000 and 41,000 people, respectively. These numbers are insufficient in comparison to the total population in the coastal region of Padang (i.e. ~200,000 people). Therefore, it is highly recommended to increase the number of TES near the coastal areas in Padang. Importantly, the TES assessment results highlight that the stochastic tsunami simulation method is able to capture the uncertainty of the future tsunamigenic impacts and hence, is essential to use this method for developing an effective tsunami mitigation plan.

### 3.3 Tsunami evacuation maps

This section presents the tsunami evacuation maps based on the stochastic tsunami inundation depths in Padang. The developed tsunami evacuation maps consist of tsunami inundation maps and horizontal, vertical, and integrated evacuation time maps to the safe zones. Three scenarios are considered to develop tsunami evacuation time maps for the $M_w$ 9.0 scenario, i.e. the 10th percentile, the 50th percentile, and the 90th percentile. Note that the tsunami evacuation time maps

developed in this section are based on the total evacuation time as presented in Section 2.2.4. The $M_w$ 9.0 scenario is chosen
because it causes the most significant tsunami impacts in Padang (see inundation maps in FIGURE 8).

FIGURE 8C shows tsunami inundation maps in Padang, whilst FIGURE 17 illustrates tsunami evacuation time maps to the safe zones in Padang for the 10th percentile (FIGURE 17A), the 50th percentile (FIGURE 17B), and the 90th percentile (FIGURE 17C). The horizontal tsunami evacuation time to the safe areas is calculated and is used to produce the tsunami evacuation time maps. The evacuation speed during the disaster event is chosen as 0.91 m/s. The total evacuation time in the 10th percentile case is sufficient to evacuate people from the coastal areas to the safe zone since the maximum evacuation time during this scenario event is about 25 minutes (see FIGURE 17A). For the 50th percentile case, some people located closer to the coastal line may need more than 30 minutes for evacuation (see FIGURE 17B). The most critical condition occurs in the case of the 90th percentile. A large population will need more than 50 minutes to evacuate to the safe zone and hence, the vertical evacuation shelters are necessary to save the people residing in those areas.

Based on the TES assessment results, the vertical and integrated evacuation time maps are further developed to investigate the possibility of reducing evacuation time to the safe areas. FIGURE 17D and FIGURE 17E show the vertical and integrated tsunami evacuation time maps to the TES locations. The shelter numbers 1, 15, 16 and 17 are excluded while vertical and integration tsunami evacuation time maps are developed because those shelters may be unsafe and inundated during the worst tsunamigenic event (the 90th percentile of the $M_w$ 9.0 scenario; see Section 3.2). In general, the vertical evacuation time map highlights that those shelters can only be accessed by the community located near the shelters (FIGURE 17D). In addition, the integrated map shows that the availability of shelters is essential to save those residents in the critical regions (see FIGURE 17E). Generally, by incorporating the vertical evacuation shelters and reducing the initial reaction time, the total evacuation time can be shorter. Therefore, besides increasing the number of TES buildings in Padang, large-scale tsunami evacuation drills in coastal community must be conducted to improve awareness of the tsunami hazard in Padang. Consequently, the casualties due to significant tsunamigenic events can be reduced.

## 4 Conclusions and outlook

The main purpose of this study was to develop effective tsunami evacuation plans in Padang based on the stochastic earthquake scenarios. Rigorous tsunami hazard assessments in Padang have been carried out using a novel stochastic tsunami simulation method to estimate the tsunami hazard level in Padang using three magnitude scenarios including $M_w$ 8.5, $M_w$ 8.75, and $M_w$ 9.0. The stochastic earthquake scenarios were generated by adopting an asperity zone from the 1797 historical event and by considering the uncertainty and dependency of earthquake source parameters. For each magnitude, 100 stochastic earthquake source scenarios (300 models in total) were generated and implemented to run the Monte Carlo tsunami simulation. The assessment of tsunami hazard in Padang was then conducted based on the stochastic tsunami inundation depth (vertical relative distance from water free-surface to ground). Subsequently, the vulnerability assessment of TES considering the effects due to ground shaking and tsunami in Padang was carried out to evaluate the adequacy during

the critical tsunami events. Finally, the hazard level assessment results were used to construct the tsunami inundation depth maps and tsunami evacuation time maps (i.e. horizontal, vertical, and integrated evacuation time maps) to better inform emergency response and rescue teams of current tsunami risks to residents in Padang. The evacuation time maps were developed for the three percentile levels (i.e. $10^{th}$, $50^{th}$, and $90^{th}$) of the $M_w$ 9.0 scenario.

For the $M_w$ 9.0 scenario, the tsunami inundation areas in Padang ranged from 4.27 km$^2$ for the $10^{th}$ percentile to 19.43 km$^2$ for the $90^{th}$ percentile with the maximum inundation depth reaching 10 m. The results clearly demonstrated that Padang may face a significant impact of the tsunami in the case of the low-probability high-consequence events. People who live near the coast will require about 60 minutes evacuating to the safe zone (i.e. to inland high ground). In such situations, resistant vertical evacuation structures should be designed and constructed in the populated areas of Padang along the coast.

The results from the seismic vulnerability assessment of the existing 23 TES in Padang indicated that all TES buildings may be usable as emergency evacuation shelter during the ground shaking event prior to tsunami. However, shelter numbers 16 and 17 are found to be unsafe during the worst tsunami event, whilst shelter numbers 1 and 15 may not be operational because all floors are inundated. Therefore, the capacity may be insufficient to accommodate a large population in the coastal region and hence, the number of TES must be increased.

Lastly, although assessments for developing evacuation plans in Padang have been conducted in this study using the results of rigorous stochastic tsunami simulations, some limitations need to be addressed in future studies. These include: (1) tsunami hazard simulations should be conducted using high-resolution DEM (e.g. 10 m), and (2) other tsunami hazard parameters, e.g. flow velocity and momentum flux, as well as other tsunami evacuation parameters, e.g. population distribution and road access, should be taken into account when assessing the adequacy of TES in Padang.

**Acknowledgments**

The first author is grateful to the Directorate General of Resources for Science, Technology and Higher Education, Ministry of Research, Technology and Higher Education of Indonesia which sponsor his PhD study. This work is also funded by the Engineering and Physical Sciences Research Council (EP/M001067/1). The authors are grateful to Sigit Sutikno who provided the TES survey data in Padang. The bathymetry and elevation data for the Sumatra region were obtained from the

GEBCO2014    database    (http://www.gebco.net/data_and_products/gridded_bathymetry_data/),    the    SRTM1 (https://lta.cr.usgs.gov/SRTM1Arc)    database    and    the    GDEM2    database    (https://asterweb.jpl.nasa.gov/gdem.asp), respectively.

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

**Figure captions**

**FIGURE 1.** (A) Significant seismic events in the Mentawai segment of the Sunda subduction zone (Sim: Simeulue, Ni: Nias, Ba: Batu, Sib: Sibereut, Sip: Sipora, Pag: Pagai Islands, Eng: Enggano). (B) Fault plane of the Mentawai segment source along with the asperity.

**FIGURE 2.** Procedures of stochastic tsunami simulation.

**FIGURE 3.** (A) Tsunami evacuation shelters in Padang. (B) Earthquake source models to study the effect of DEM on tsunami simulation.

**FIGURE 4.** Digital Elevation Model for Padang: (A) DEM5. (B) SRTM1. (C) GDEM2.

**FIGURE 5.** Elevation differences of global DEM datasets with respect to DEM5: (A) SRTM1. (B) GDEM2.

**FIGURE 6.** (A) Capacity spectrum method. (B). HAZUS-seismic fragility curves. (C). Tsunami fragility curves developed by Suppasri et al. (2011)

**FIGURE 7.** Inundation areas in Padang for: (A) DEM5. (B) SRTM1. (C) GDEM2 (IA = total inundation areas).

**FIGURE 8.** Inundation depth maps in Padang: (A) $M_w$ 8.5 scenario. (B) $M_w$ 8.75 scenario. (C) $M_w$ 9.0 scenario (IA = inundation area).

**FIGURE 9.** Inundation areas above 1 m depth in Padang: (A) $M_w$ 8.5 scenario. (B) $M_w$ 8.75 scenario. (C) $M_w$ 9.0 scenario (IA = inundation area).

**FIGURE 10.** (A) Site location. (B) Maximum tsunami inundation height in the coastal line of Padang for the $M_w$ 8.5 scenario. (C) Maximum tsunami inundation height in the coastal line of Padang for the $M_w$ 8.75 scenario. (D) Maximum tsunami inundation height in the coastal line of Padang for the $M_w$ 9.0 scenario.

**FIGURE 11.** (A) Site location. (B) Maximum tsunami inundation height in the first river of Padang for the $M_w$ 8.5 scenario. (C) Maximum tsunami inundation height in the first river of Padang for the $M_w$ 8.75 scenario. (D) Maximum tsunami inundation height in the first river of Padang for the $M_w$ 9.0 scenario.

**FIGURE 12.** (A) Site location. (B) Maximum tsunami inundation height in the second river of Padang for the $M_w$ 8.5 scenario. (C) Maximum tsunami inundation height in the second river of Padang for the $M_w$ 8.75 scenario. (D) Maximum tsunami inundation height in the second river of Padang for the $M_w$ 9.0 scenario.

**FIGURE 13.** (A) Site location. (B) Maximum tsunami inundation height in the third river of Padang for the $M_w$ 8.5 scenario. (C) Maximum tsunami inundation height in the second river of Padang for the $M_w$ 8.75 scenario. (D) Maximum tsunami inundation height in the third river of Padang for the $M_w$ 9.0 scenario.

**FIGURE 14.** Inundation depth variability at TES stations for (A) $M_w$ 8.5 scenario. (B) $M_w$ 8.75 scenario. (C) $M_w$ 9.0 scenario.

**FIGURE 15.** (A-K) Inundation depth variability at TES stations located near the coastal line and the rivers.

**FIGURE 16.** (A-C). The seismic vulnerability assessment of TES for the worst scenario using the HAZUS methodology. (D) Simulated spectral acceleration from the 100 tsunamigenic scenarios. (E) Simulated peak ground acceleration from the 100 tsunamigenic scenarios (F) The probability exceedance of the three damage states from the 100 earthquake scenarios.

**FIGURE 17.** (A) Horizontal tsunami evacuation time maps in Padang for the 10[th] percentile. (B) Horizontal tsunami evacuation time maps in Padang for the 50[th] percentile. (C) Horizontal tsunami evacuation time maps in Padang for the 90[th] percentile. (D) Vertical tsunami evacuation time maps in Padang for the 90[th] percentile. (E) Integrated tsunami evacuation time maps in Padang for the 90[th] percentile.

Table 1. Statistics of elevation differences between global DEM datasets (i.e. SRTM1 and GDEM2) and the reference data (DEM5).

| DEM | Min (m) | Max (m) | Mean difference (m) | Absolute mean difference (m) | RMSE (m) |
|---|---|---|---|---|---|
| DEM5 | 0 | 280 | - | - | - |
| SRTM1 | 0 | 273 | 1.48 | 3.13 | 4.27 |
| GDEM2 | 0 | 268 | 3.96 | 5.69 | 7.46 |

Table 2. Tsunami evacuation shelters in Padang.

| No. | Name of the shelter | Capacity (persons) | Location Longitude | Location Latitude | Maximum height (m) | Number of floors | Evacuation area (m$^2$) |
|---|---|---|---|---|---|---|---|
| 1 | Sport centre of Universitas Negeri Padang (UNP) | 1,500 | 100.3474 | -0.89979 | 10 | 1 | 2475 |
| 2 | Art building of UNP | 2,000 | 100.3488 | -0.89998 | 20 | 5 | 3300 |
| 3 | DPRD province building | 2,000 | 100.3515 | -0.90628 | 15 | 3 | 3300 |
| 4 | Post-graduate building of Universitas Bung Hatta (UBH) | 2,000 | 100.3434 | -0.90677 | 15 | 4 | 3300 |
| 5 | Al-Azhar primary school | 1,100 | 100.3544 | -0.90924 | 10 | 3 | 1815 |
| 6 | BPK office of West Sumatra | 2,000 | 100.3566 | -0.91127 | 20 | 4 | 3300 |
| 7 | Office of KANWIL DITJEN perbendaharaan | 2,000 | 100.3587 | -0.9161 | 15 | 3 | 3300 |
| 8 | Senior High School 1 of Padang | 1,400 | 100.3539 | -0.91923 | 10 | 3 | 2310 |
| 9 | Junior High School 25 of Padang | 1,000 | 100.3568 | -0.92025 | 10 | 3 | 1650 |
| 10 | Senior Vocational High School 5 of Padang | 3,000 | 100.3519 | -0.92178 | 10 | 3 | 4950 |
| 11 | Grand Mosque of West Sumatra | 4,000 | 100.3625 | -0.92423 | 47 | 2 | 6600 |
| 12 | BAPPEDA province office | 1,500 | 100.3609 | -0.92589 | 15 | 3 | 2475 |
| 13 | Ibis hotel of Padang | 3,000 | 100.3629 | -0.9294 | 52 | 13 | 4950 |
| 14 | PrasJal office of West Sumatra Province | 5,000 | 100.3637 | -0.92953 | 15 | 4 | 8250 |
| 15 | Elementary School 24 of Purus | 3,000 | 100.3546 | -0.93371 | 10 | 3 | 4950 |
| 16 | Mercure Hotel of Padang | 4,000 | 100.3527 | -0.9359 | 30 | 8 | 6600 |
| 17 | RUSUNAWA | 3,200 | 100.3516 | -0.93681 | 15 | 5 | 5280 |
| 18 | Governor office of West Sumatra | 3,500 | 100.3606 | -0.94116 | 20 | 4 | 5775 |
| 19 | Office of Bank Indonesia | 1,000 | 100.3623 | -0.94336 | 10 | 2 | 1650 |
| 20 | Nurul Haq mosque | 4,000 | 100.3536 | -0.95091 | 22 | 5 | 6600 |
| 21 | Grand Zuri Padang Hotel | 3,000 | 100.3641 | -0.95467 | 25 | 6 | 4950 |
| 22 | Nurul Iman mosque | 1,000 | 100.3623 | -0.95473 | 10 | 2 | 1650 |
| 23 | Grand Inna Muara Hotel | 4,000 | 100.357 | -0.95734 | 25 | 6 | 6600 |

Table 3. Description of seismic damage state (HAZUS, 2003; Rosetto and Elnashai, 2003)

| Damage state | | Description |
|---|---|---|
| DS1 | Slight | Ranging from minimal damage of the structural and non-structural components to minor nonstructural and structural damage |
| DS2 | Moderate | Moderate non-structural and structural damage |
| DS3 | Extensive | Extensive structural and non-structural damage. Localised life threatening situations are common. |
| DS4 | Complete (collapse) | Building is fully destroyed, with significant portions of the building collapsed. |

Table 4. Damage state definitions for tsunami (Suppasri et al., 2014; Charvet et al., 2017).

| Damage state | | Description | Use |
|---|---|---|---|
| DST0 | No damage | Water does not flow into the building | Immediate occupancy |
| DST1 | Minor damage | Water enters below the ground floor | Possible to use immediately after minor clean-up |
| DST2 | Moderate damage | Water inundates to less than 1 m above the ground floor | Possible to use immediately after moderate repairs |
| DST3 | Major damage | Water inundates to more than 1 m above the ground floor (but below the ceiling | Possible to use immediately after major repairs |
| DST4 | Complete damage | The building is inundated above the ground floor level | Major repairmen is needed |
| DST5 | Collapsed | Structural elements are significantly damage | Not repairable |
| DST6 | Washed away | The building is completely washed away | Not repairable |

Table 5. Assessment of tsunami evacuation shelters (TES) in Padang in terms of depth, tsunami arrival time, and number of tsunami
destructive events.

| No. | TES height (m) | 10th percentile Depth (m) | 50th percentile Depth (m) | 90th percentile | | Percentage of destructive events (%) |
|---|---|---|---|---|---|---|
| | | | | Depth (m) | Tsunami arrival time (minutes) | |
| 1 | 10 | 0 | 2.9 | 6 | 23 | 16 |
| 2 | 20 | 0.5 | 3.9 | 7 | 24 | 25 |
| 3 | 15 | 0 | 1.4 | 4.6 | 25 | 6 |
| 4 | 15 | 0 | 2.6 | 5.7 | 19 | 11 |
| 5 | 10 | 0 | 0.1 | 3.1 | 27 | 1 |
| 6 | 20 | 0 | 0 | 2.5 | 30 | 1 |
| 7 | 15 | 0 | 0 | 1.9 | 30 | 1 |
| 8 | 10 | 0 | 0.8 | 3.6 | 24 | 5 |
| 9 | 10 | 0 | 0.2 | 3 | 25 | 1 |
| 10 | 10 | 0 | 2.1 | 5.1 | 21 | 8 |
| 11 | 47 | 0 | 0 | 2.2 | 30 | 1 |
| 12 | 15 | 0 | 0 | 2.8 | 28 | 1 |
| 13 | 52 | 0 | 0.8 | 4 | 28 | 4 |
| 14 | 15 | 0 | 0 | 3.1 | 30 | 1 |
| 15 | 10 | 0.3 | 3.6 | 6.7 | 20 | 24 |
| 16 | 30 | 1.2 | 4.3 | 7.5 | 17 | 30 |
| 17 | 15 | 1.6 | 4.8 | 7.8 | 15 | 36 |
| 18 | 20 | 0 | 0.5 | 3.8 | 27 | 1 |
| 19 | 10 | 0 | 0 | 3 | 29 | 1 |
| 20 | 22 | 1.3 | 4 | 6.8 | 17 | 25 |
| 21 | 25 | 0 | 0.3 | 2.9 | 28 | 1 |
| 22 | 10 | 0 | 0.5 | 3.1 | 27 | 1 |
| 23 | 25 | 0.5 | 2.9 | 5.6 | 20 | 12 |

Table 6. TES building capacity during the tsunami event considering the Mw 9.0 scenario.

| No. | Building height | Total floor | Total evacuation area (m²) | Evacuation area in each floor (m²) | 10th percentile | | | 50th percentile | | | 90th percentile | | |
|---|---|---|---|---|---|---|---|---|---|---|---|---|---|
| | | | | | Inundation depth (m) | Number of evacuation floor | Capacity during tsunami (person) | Inundation depth (m) | Number of evacuation floor | Capacity during tsunami (person) | Inundation depth (m) | Number of evacuation floor | Capacity during tsunami (person) |
| 1 | 10 | 1 | 2475 | 2475 | 0 | 1 | 2475 | 2.9 | 0 | 0 | 6 | 0 | 0 |
| 2 | 20 | 5 | 3300 | 660 | 0.5 | 4 | 2640 | 3.9 | 3 | 1980 | 7 | 2 | 1320 |
| 3 | 15 | 3 | 3300 | 1100 | 0 | 3 | 3300 | 1.4 | 2 | 2200 | 4.6 | 1 | 1100 |
| 4 | 15 | 4 | 3300 | 825 | 0 | 4 | 3300 | 2.6 | 3 | 2475 | 5.7 | 2 | 1650 |
| 5 | 10 | 3 | 1815 | 605 | 0 | 3 | 1815 | 0.1 | 2 | 1210 | 3.1 | 1 | 605 |
| 6 | 20 | 4 | 3300 | 825 | 0 | 4 | 3300 | 0 | 4 | 3300 | 2.5 | 3 | 2475 |
| 7 | 15 | 3 | 3300 | 1100 | 0 | 3 | 3300 | 0 | 3 | 3300 | 1.9 | 2 | 2200 |
| 8 | 10 | 3 | 2310 | 770 | 0 | 3 | 2310 | 0.8 | 2 | 1540 | 3.6 | 1 | 770 |
| 9 | 10 | 3 | 1650 | 550 | 0 | 3 | 1650 | 0.2 | 2 | 1100 | 3 | 2 | 1100 |
| 10 | 10 | 3 | 4950 | 1650 | 0 | 3 | 4950 | 2.1 | 2 | 3300 | 5.1 | 1 | 1650 |
| 11 | 47 | 2 | 6600 | 3300 | 0 | 2 | 6600 | 0 | 2 | 6600 | 2.2 | 1 | 3300 |
| 12 | 15 | 3 | 2475 | 825 | 0 | 3 | 2475 | 0 | 3 | 2475 | 2.8 | 2 | 1650 |
| 13 | 52 | 13 | 4950 | 381 | 0 | 13 | 4950 | 0.8 | 12 | 4569 | 4 | 11 | 4188 |
| 14 | 15 | 4 | 8250 | 2063 | 0 | 4 | 8250 | 0 | 4 | 8250 | 3.1 | 2 | 4125 |
| 15 | 10 | 3 | 4950 | 1650 | 0.3 | 2 | 3300 | 3.6 | 1 | 1650 | 6.7 | 0 | 0 |
| 16 | 30 | 8 | 6600 | 825 | UNSAFE | | | | | | | | |
| 17 | 15 | 5 | 5280 | 1056 | | | | | | | | | |
| 18 | 20 | 4 | 5775 | 1444 | 0 | 4 | 5775 | 0.5 | 3 | 4331 | 3.8 | 2 | 2888 |
| 19 | 10 | 2 | 1650 | 825 | 0 | 2 | 1650 | 0 | 2 | 1650 | 3 | 1 | 825 |
| 20 | 22 | 5 | 6600 | 1320 | 1.3 | 4 | 5280 | 4 | 3 | 3960 | 6.8 | 2 | 2640 |
| 21 | 25 | 6 | 4950 | 825 | 0 | 6 | 4950 | 0.3 | 5 | 4125 | 2.9 | 5 | 4125 |
| 22 | 10 | 2 | 1650 | 825 | 0 | 2 | 1650 | 0.5 | 1 | 825 | 3.1 | 1 | 825 |
| 23 | 25 | 6 | 6600 | 1100 | 0.5 | 5 | 5500 | 2.9 | 5 | 5500 | 5.6 | 4 | 4400 |
| Total capacity | | | | | | | 79420 | | | 64340 | | | 41836 |

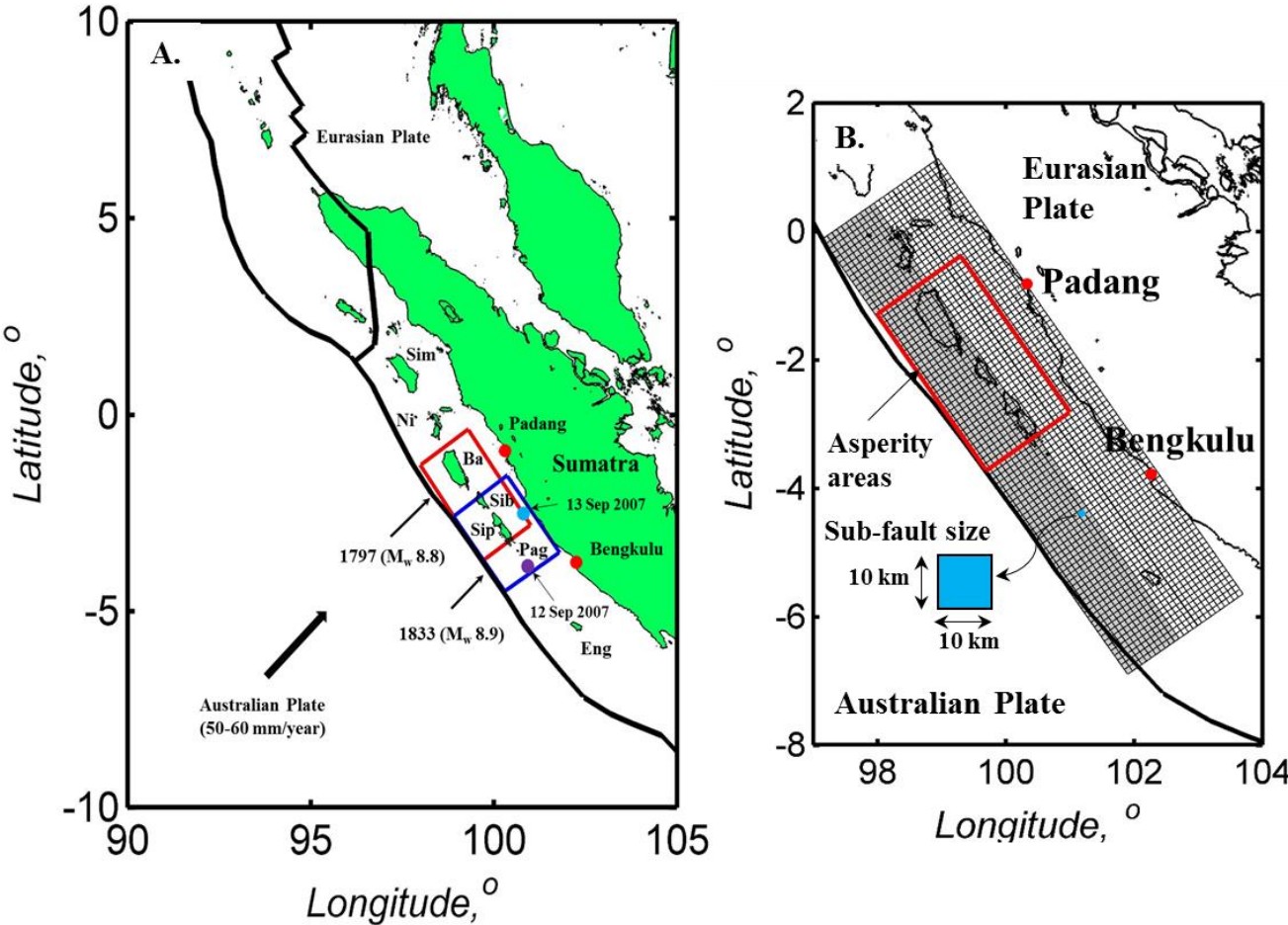

**FIGURE 1.** (A) Significant seismic events in the Mentawai segment of the Sunda subduction zone (Sim: Simeulue, Ni: Nias, Ba: Batu, Sib: Sibereut, Sip: Sipora, Pag: Pagai Islands, Eng: Enggano). (B) Fault plane of the Mentawai segment source along with the asperity.

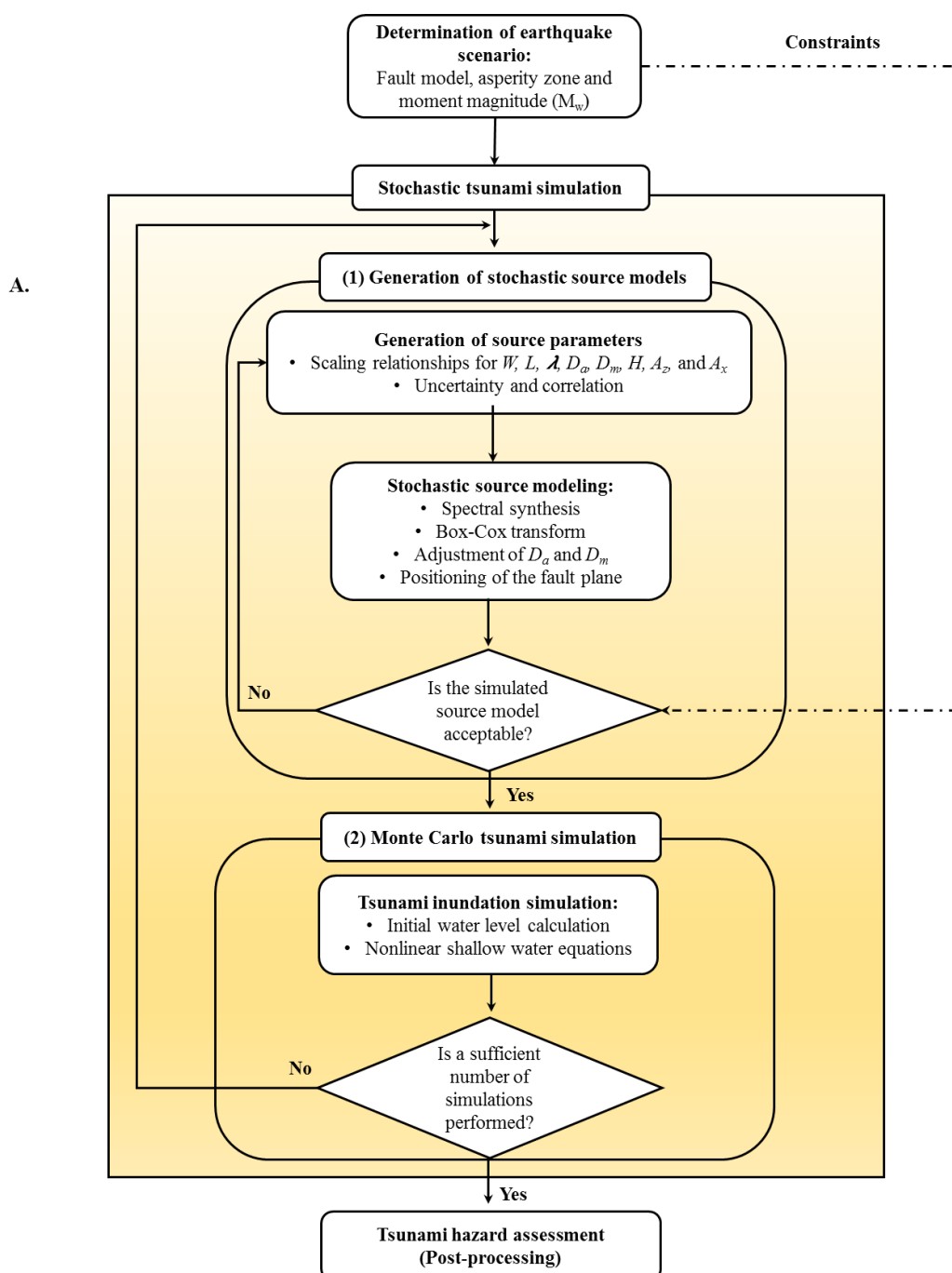

**FIGURE 2.** Procedures of stochastic tsunami simulation.

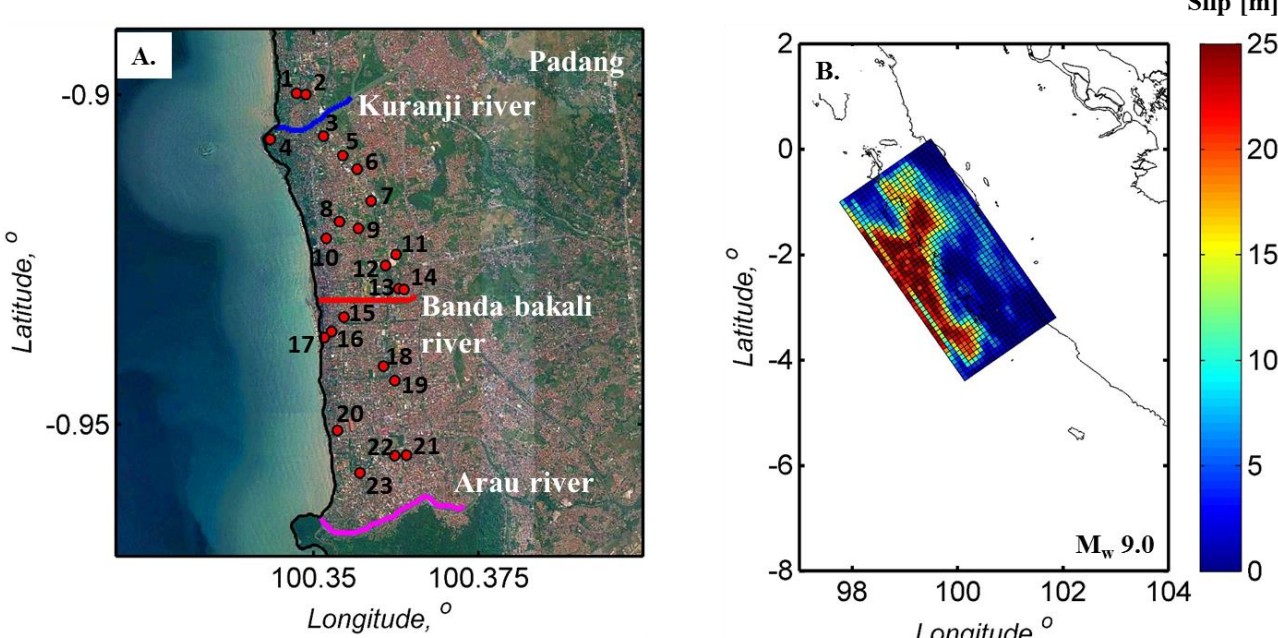

**FIGURE 3.** (A) Tsunami evacuation shelters in Padang. (B) Earthquake source models to study the effect of DEM on tsunami simulation.

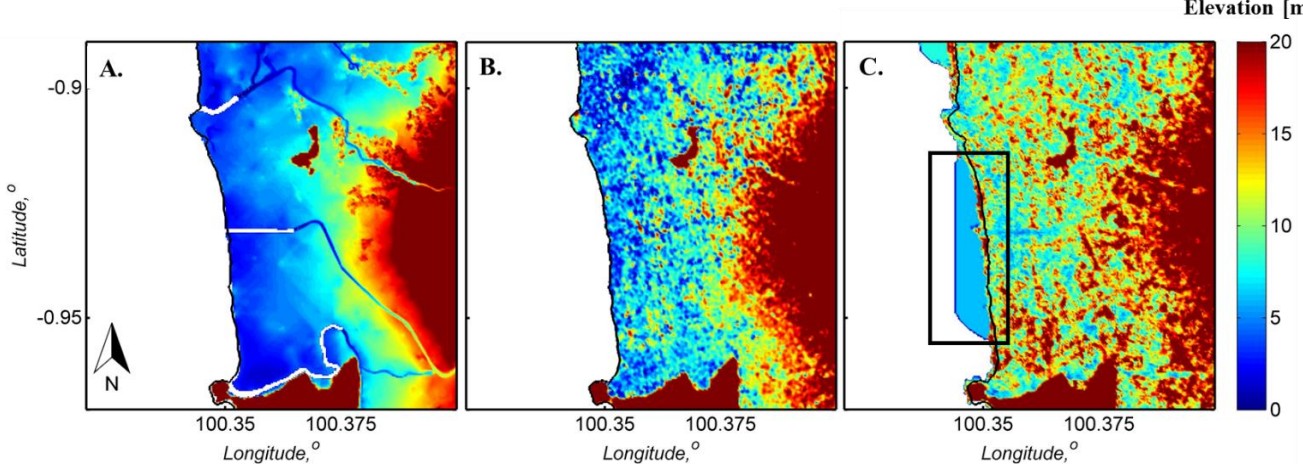

**FIGURE 4.** Digital Elevation Model for Padang: (A) DEM5. (B) SRTM1. (C) GDEM2.

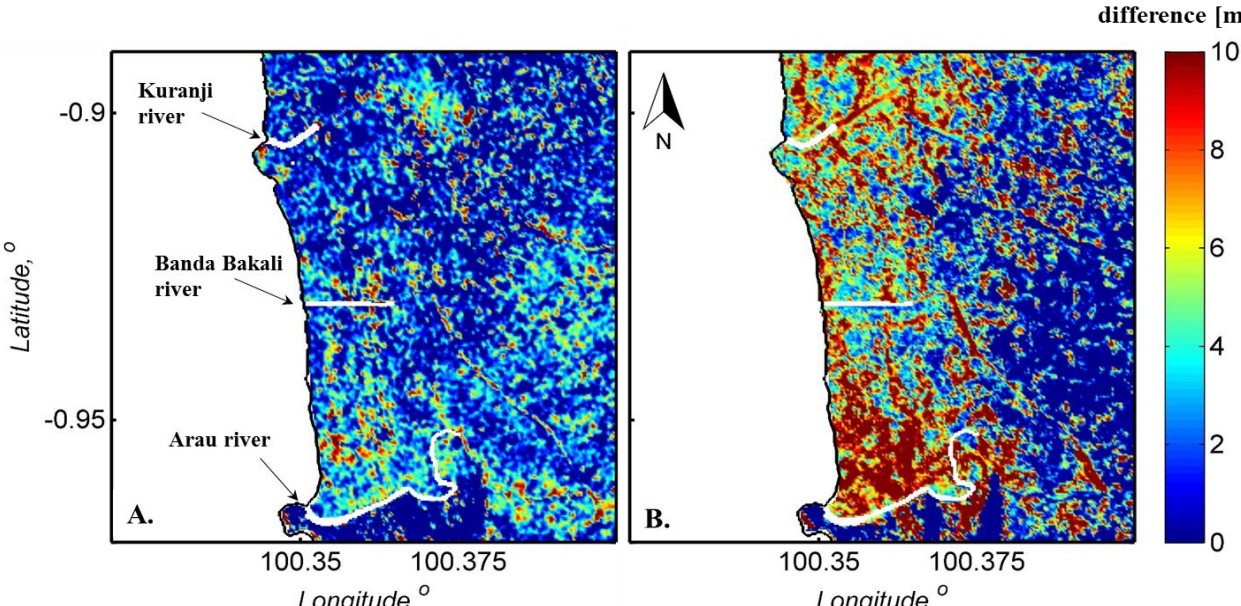

**FIGURE 5.** Elevation differences of global DEM datasets with respect to DEM5: (A) SRTM1. (B) GDEM2.

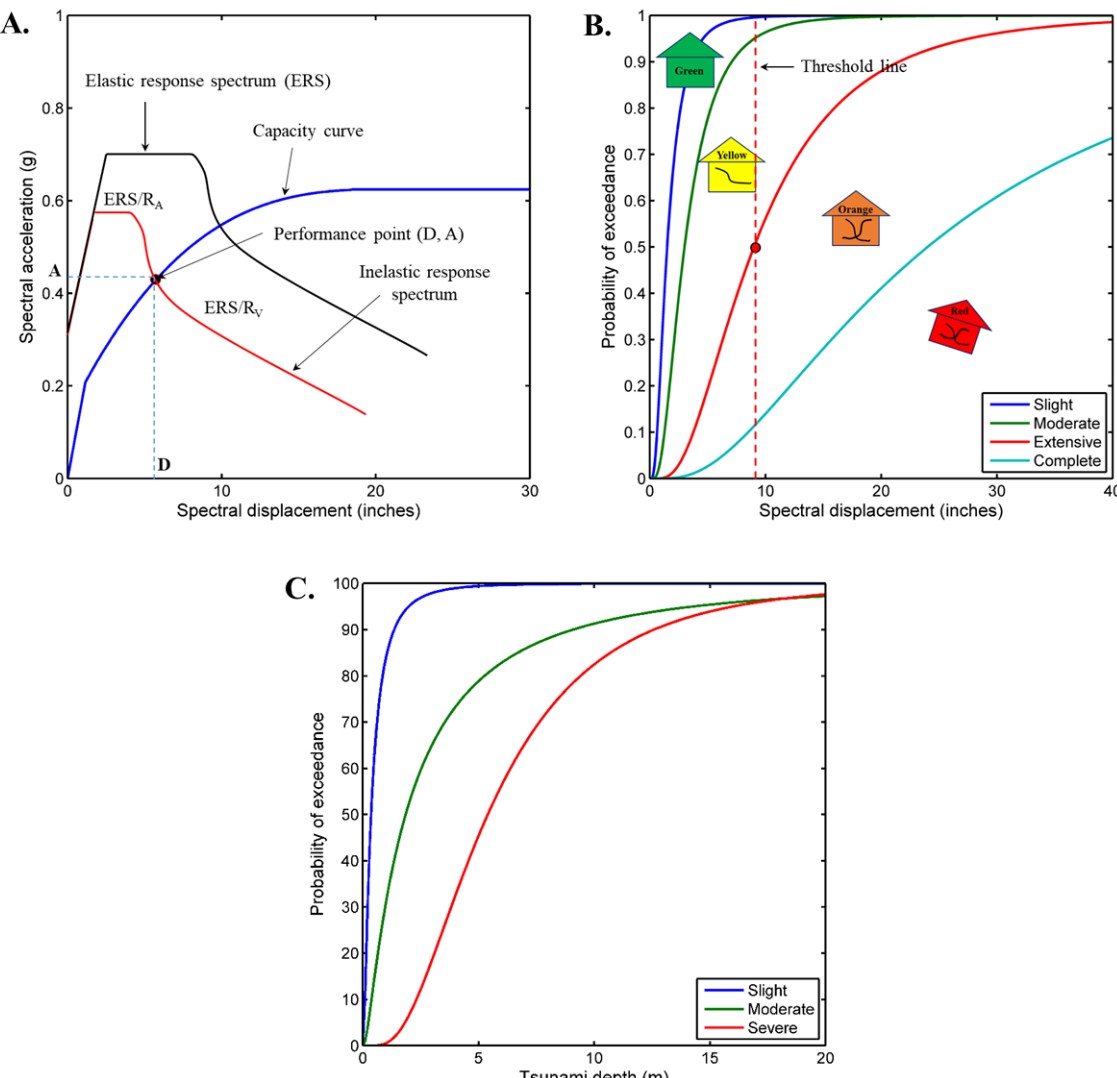

**FIGURE 6.** (A) Capacity spectrum method. (B). HAZUS-seismic fragility curves. (C). Tsunami fragility curves developed by Suppasri et
al. (2011)

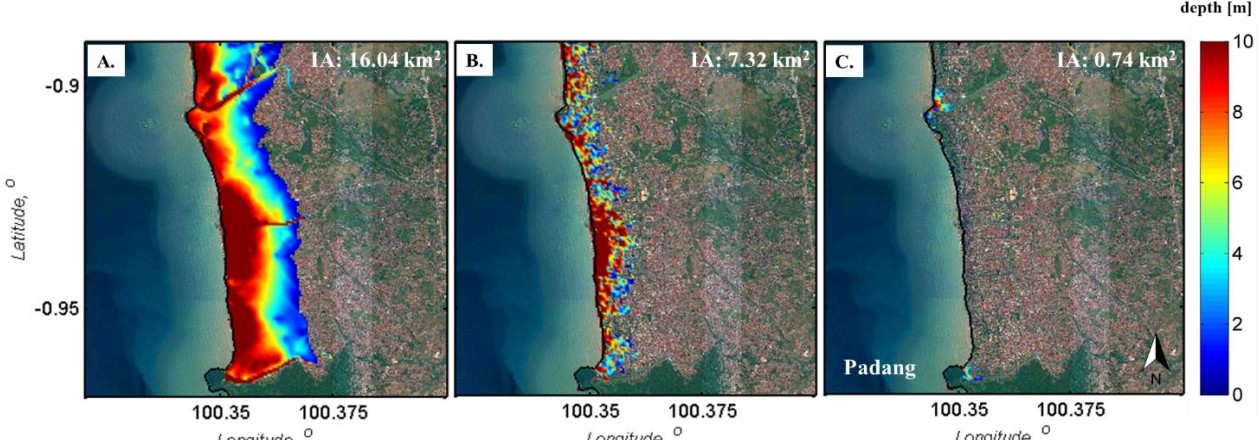

**FIGURE 7.** Inundation areas in Padang for: (A) DEM5. (B) SRTM1. (C) GDEM2 (IA = total inundation areas).

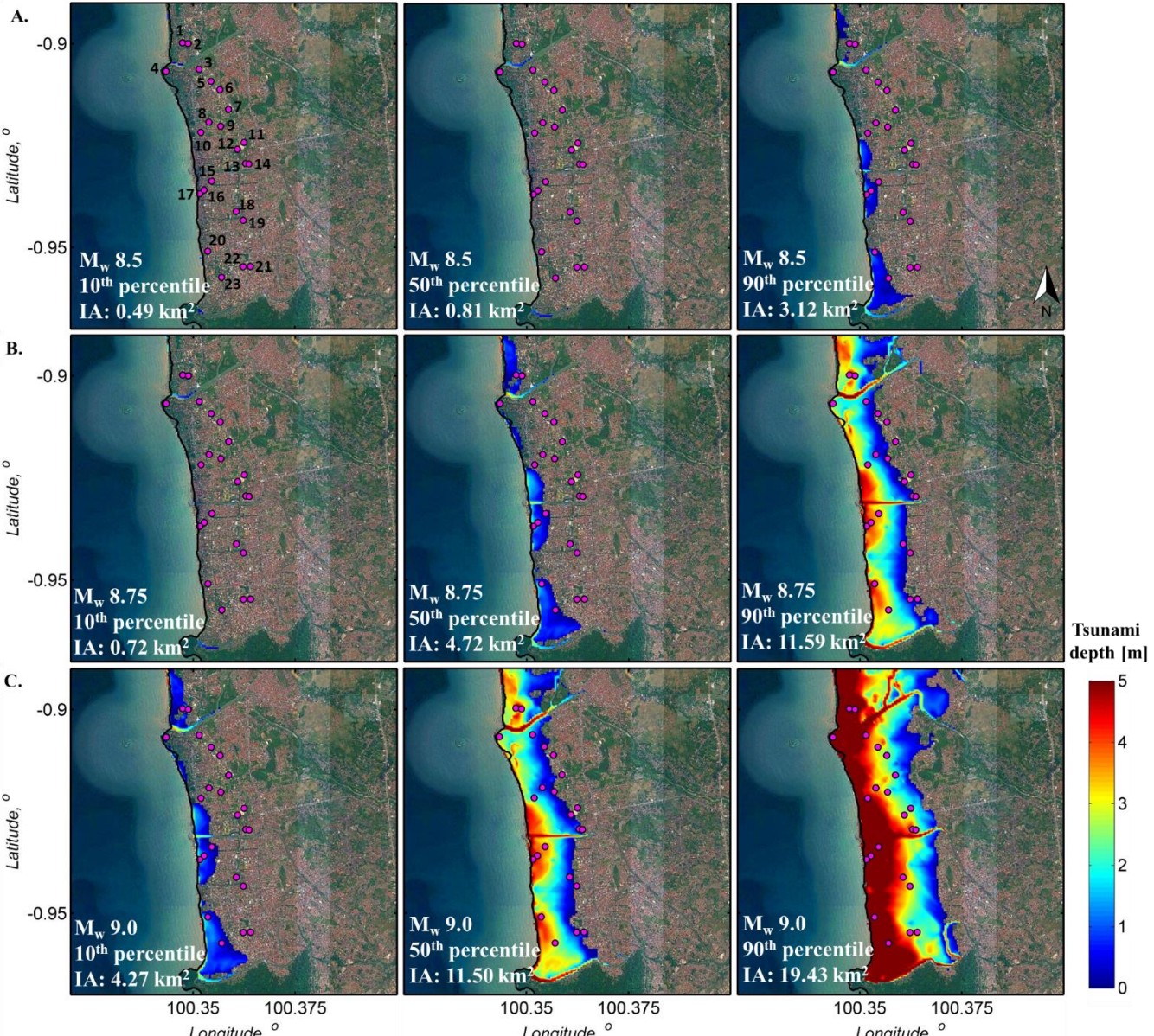

**FIGURE 8.** Inundation depth maps in Padang: (A) $M_w$ 8.5 scenario. (B) $M_w$ 8.75 scenario. (C) $M_w$ 9.0 scenario (IA = inundation area).

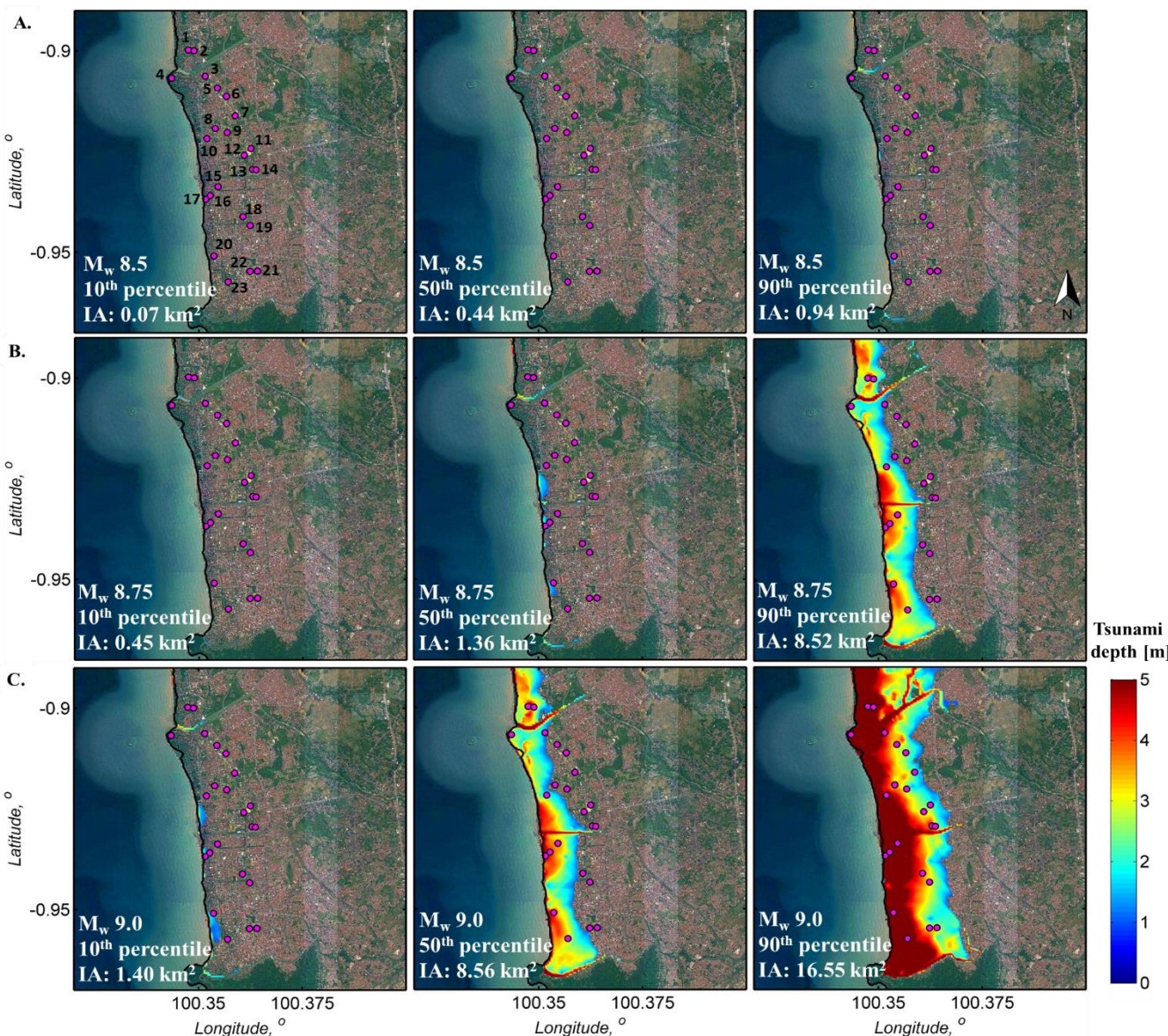

**FIGURE 9.** Inundation areas above 1 m depth in Padang: (A) $M_w$ 8.5 scenario. (B) $M_w$ 8.75 scenario. (C) $M_w$ 9.0 scenario (IA = inundation area).

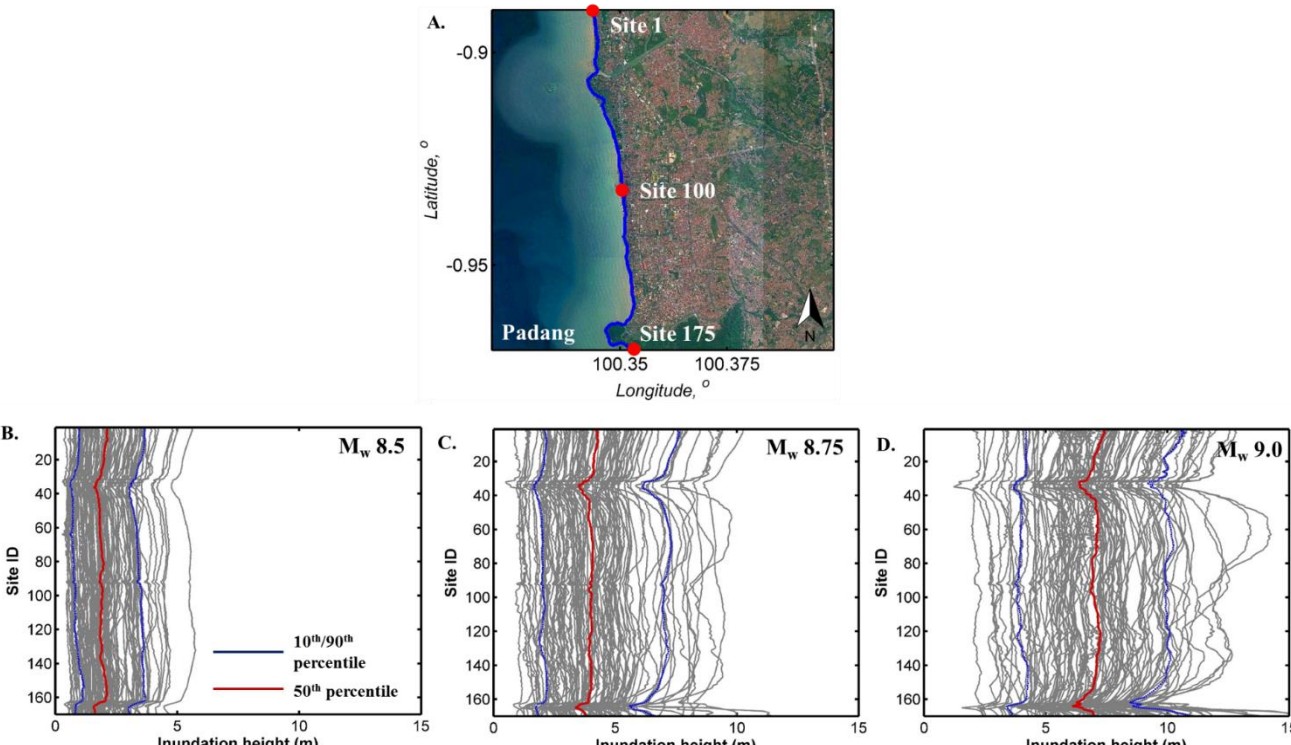

**FIGURE 10.** (A) Site location. (B) Maximum tsunami inundation height in the coastal line of Padang for the $M_w$ 8.5 scenario. (C) Maximum tsunami inundation height in the coastal line of Padang for the $M_w$ 8.75 scenario. (D) Maximum tsunami inundation height in the coastal line of Padang for the $M_w$ 9.0 scenario.

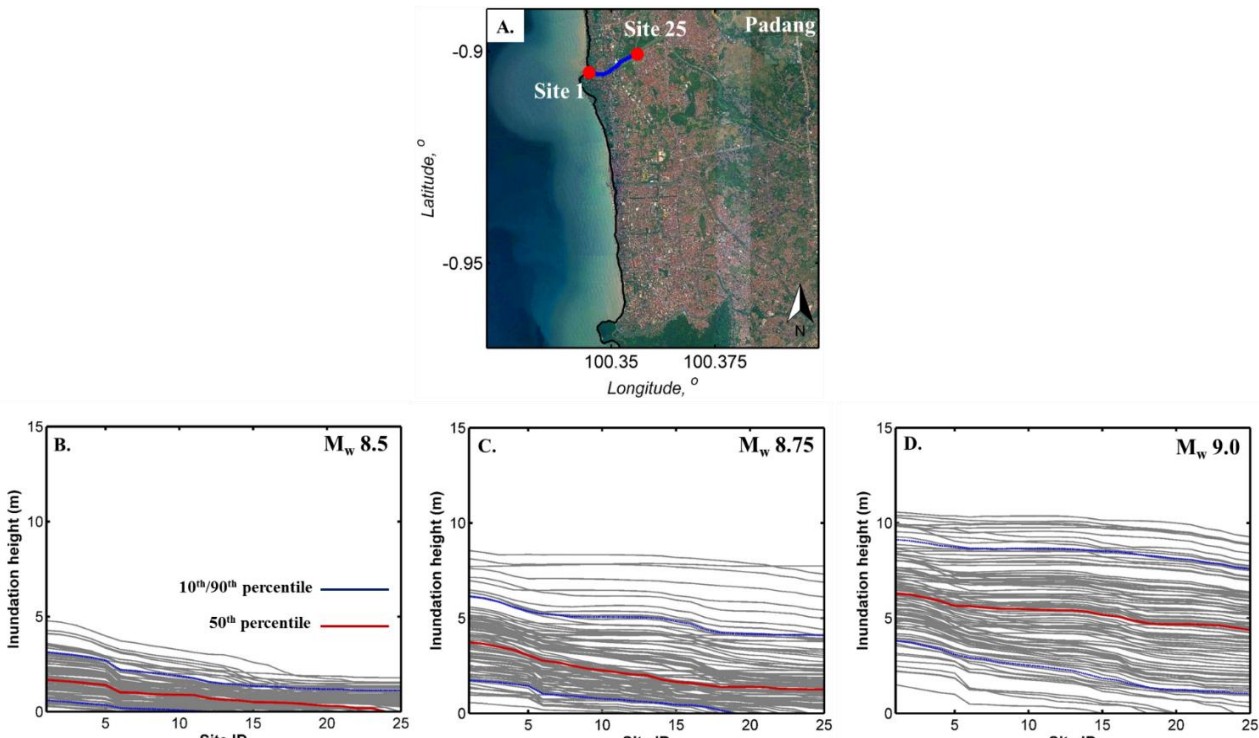

FIGURE 11. (A) Site location. (B) Maximum tsunami inundation height in the first river of Padang for the $M_w$ 8.5 scenario. (C) Maximum tsunami inundation height in the first river of Padang for the $M_w$ 8.75 scenario. (D) Maximum tsunami inundation height in the first river of Padang for the $M_w$ 9.0 scenario.

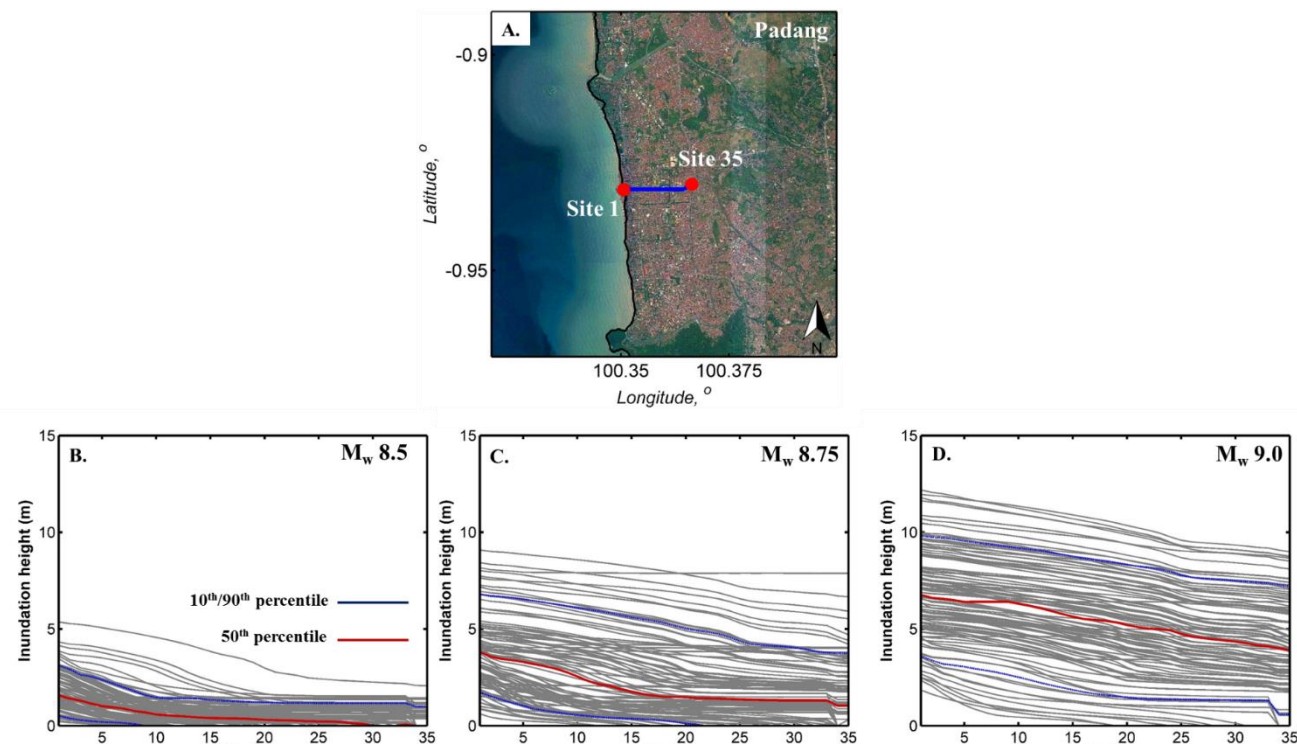

**FIGURE 12.** (A) Site location. (B) Maximum tsunami inundation height in the second river of Padang for the $M_w$ 8.5 scenario. (C)

Maximum tsunami inundation height in the second river of Padang for the $M_w$ 8.75 scenario. (D) Maximum tsunami inundation height in

the second river of Padang for the $M_w$ 9.0 scenario.

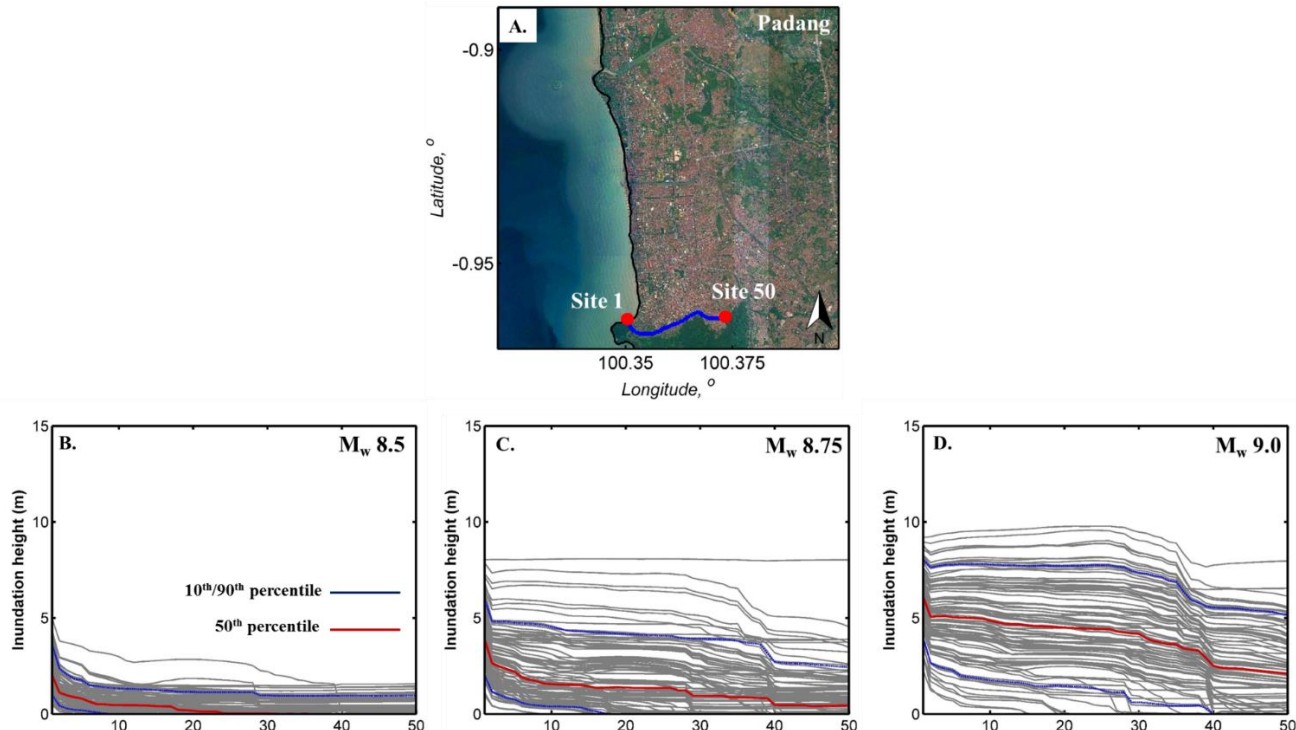

**FIGURE 13.** (A) Site location. (B) Maximum tsunami inundation height in the third river of Padang for the $M_w$ 8.5 scenario. (C)

Maximum tsunami inundation height in the second river of Padang for the $M_w$ 8.75 scenario. (D) Maximum tsunami inundation height in

the third river of Padang for the $M_w$ 9.0 scenario.

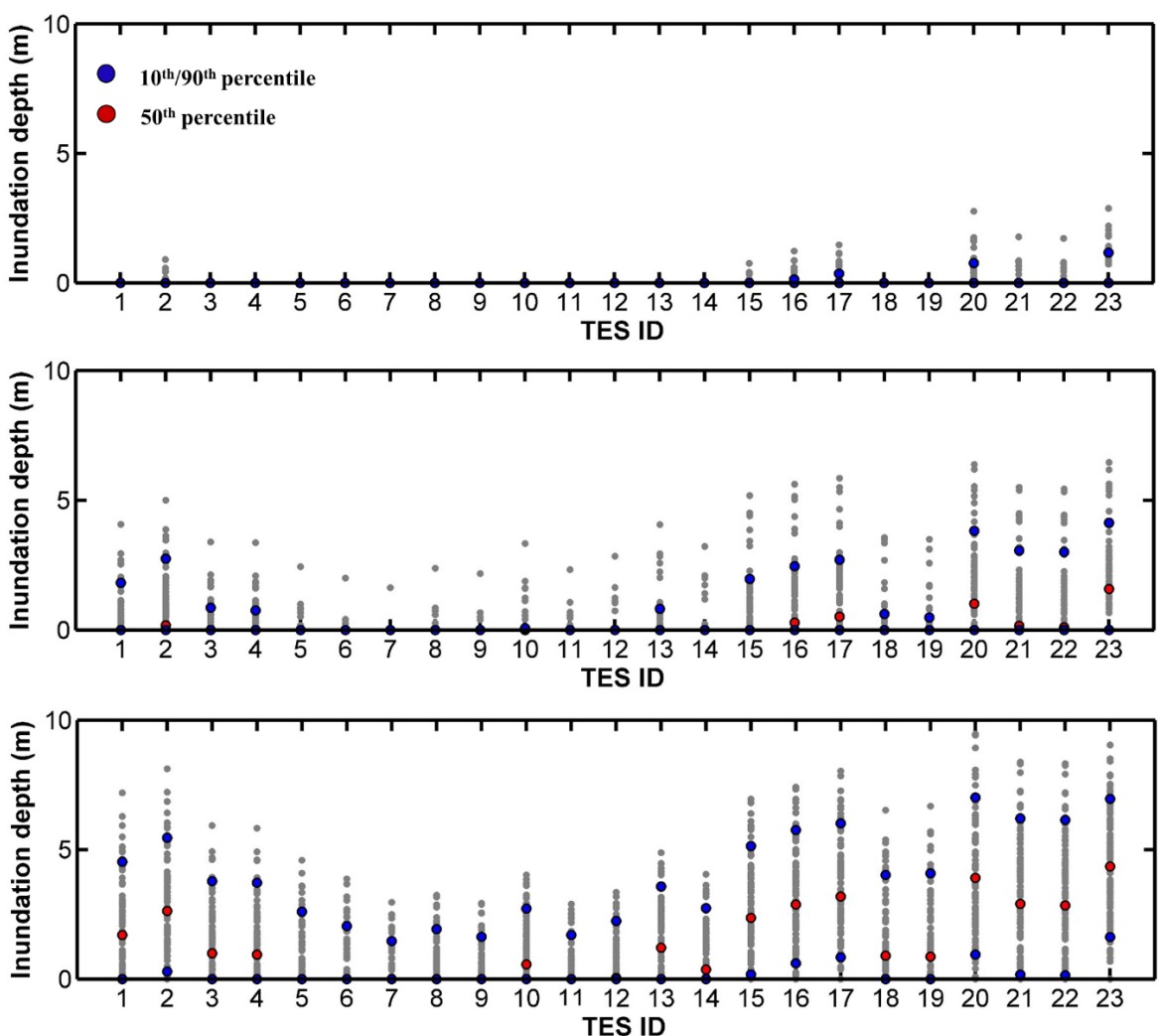

FIGURE 14. Inundation depth variability at TES stations for (A) $M_w$ 8.5 scenario. (B) $M_w$ 8.75 scenario. (C) $M_w$ 9.0 scenario.

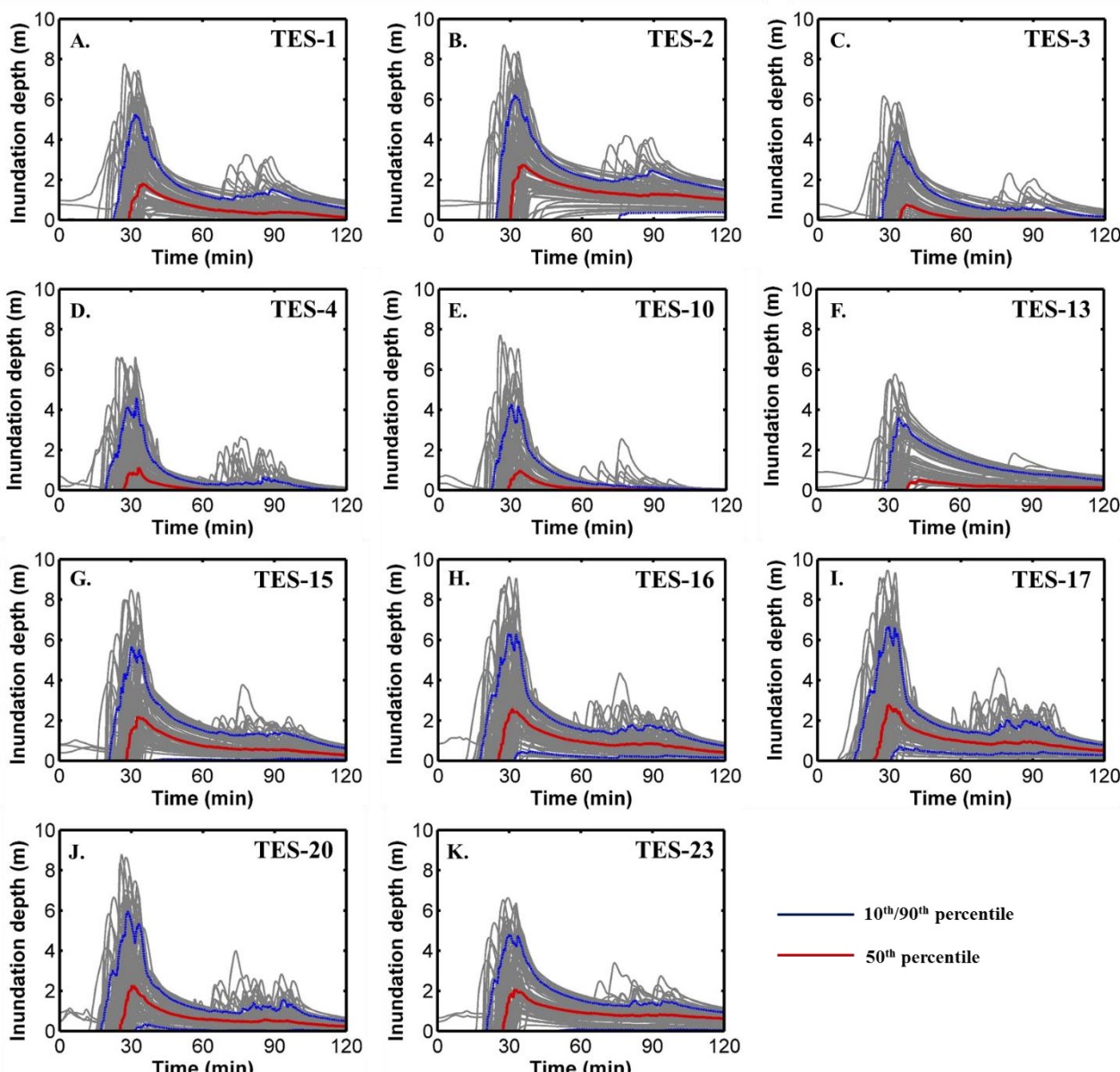

**FIGURE 15.** (A-K) Inundation depth variability at TES stations located near the coastal line and the rivers.

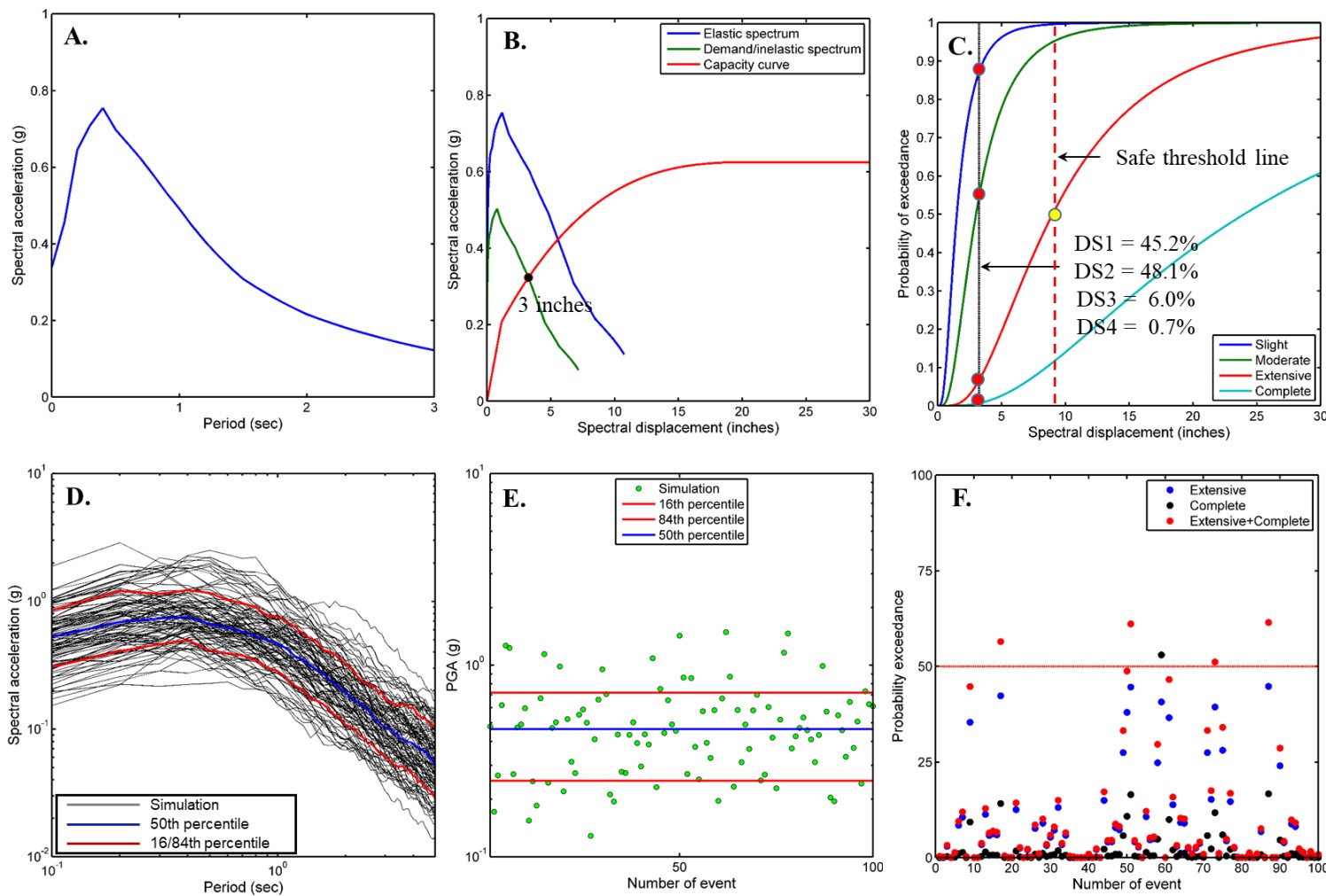

**FIGURE 16.** (A-C). The seismic vulnerability assessment of TES for the worst scenario using the HAZUS methodology. (D) Simulated spectral acceleration
from the 100 tsunamigenic scenarios. (E) Simulated peak ground acceleration from the 100 tsunamigenic scenarios (F) The probability exceedance of the three
damage states from the 100 earthquake scenarios.

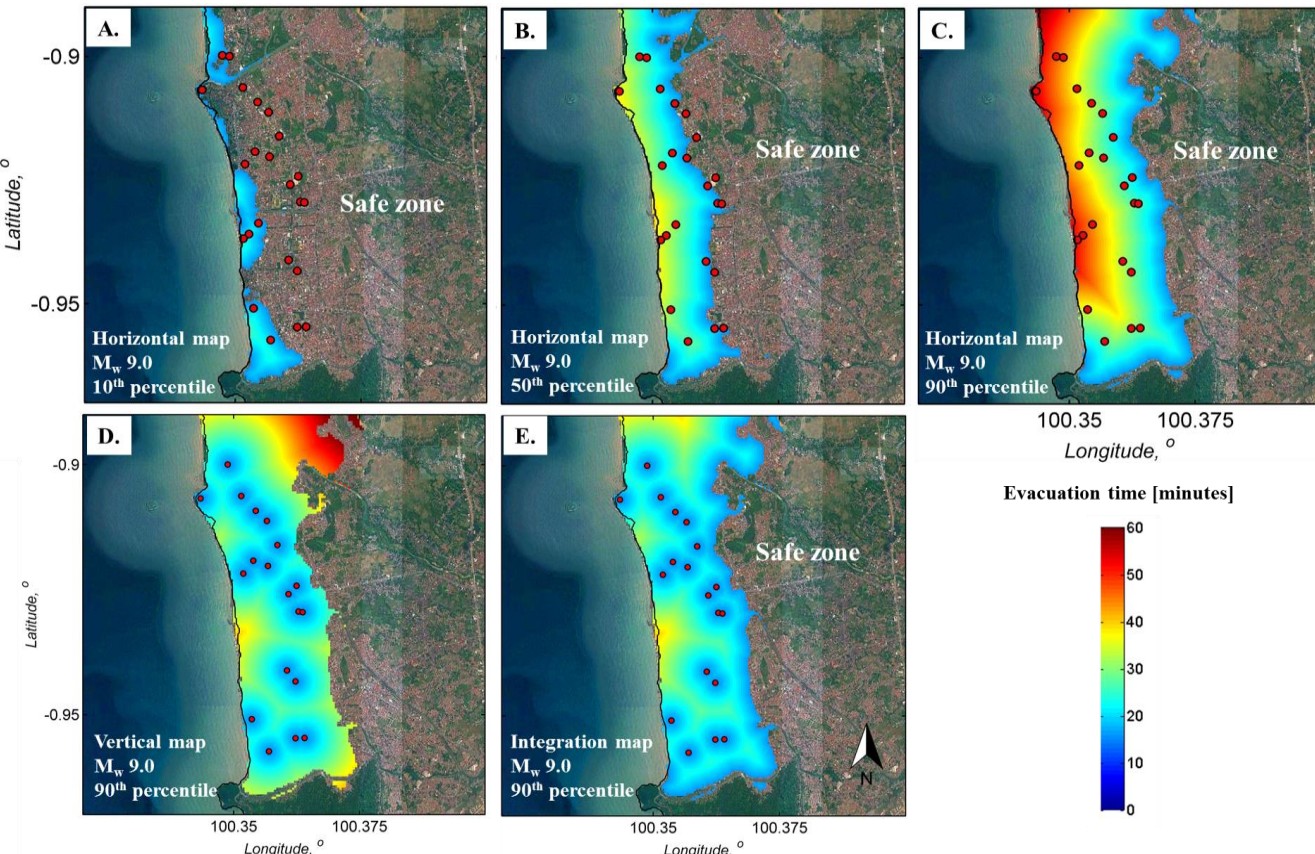

**FIGURE 17.** (A) Horizontal tsunami evacuation time maps in Padang for the 10th percentile. (B) Horizontal tsunami evacuation time maps in Padang for the 50th percentile. (C) Horizontal tsunami evacuation time maps in Padang for the 90th percentile. (D) Vertical tsunami evacuation time maps in Padang for the 90th percentile. (E) Integrated tsunami evacuation time maps in Padang for the 90th percentile.