# Peer review of "Tsunami evacuation plans for future megathrust earthquakes in Padang, Indonesia considering stochastic earthquake scenarios"

_Natural Hazards and Earth System Sciences, 2017_

## Referee Comment (RC1) · Anonymous Referee #1 · 26 Jun 2017

Comments on the manuscript nhess-2017-75 "Tsunami evacuation plans for future megathrust earthquakes in Padang, Indonesia considering stochastic earthquake scenarios" by Muhammad and co-authors

General Comments The paper by Muhammad et al. addresses the development of tsunami plans in Padang, Indonesia, using stochastic earthquake scenarios and high-resolution inundation numerical modelling. While the MS is of interest as it contributes to the improvement of the mitigation policies in a region highly vulnerable to tsunami, I find that their methodology suffers a major limitation of not addressing the building vulnerability to both earthquake and tsunami effects. Moreover, some points need to

be carefully improved, clarified and justified. In terms of the MS structure and writing, the paper is of good quality; it is well structured and is easily readable. Only the style and the grammar of some few sentences need to be improved. In terms of scientific content, in order to get the paper acceptable for publication in NHESS journal, the following comments should be carefully considered:

Specific Comments by Sections 1. INTRODUCTION: In the introduction section, there is a lack of a description of the effects of the 26th December 2004 Indian Ocean Tsunami on the study area. Therefore, I suggest that the authors add a short paragraph in which they are invited to outline the reasons why the study area of Padang, Indonesia, escaped the destructive effects of the 26 December 2004 Indian Ocean tsunami. 2. EARTHQUAKE SCENARIO SELECTION: The authors must justify their choice as regards the magnitudes of the earthquake scenarios. Why a minimum Magnitude of 8.5 and not 8.25, for instance? Is a Mw8.25 earthquake causes a tsunami with no significant effect on the study area? Why do the authors not consider a maximum earthquake magnitude in the range of this of the 2004 Indian Ocean event (Mw9.2)? Such a choice would change the predicted tsunami inundation characteristics and therefore the associated evacuation plan. 3. STOCHASTIC TSUNAMI SIMULATION: . It is not clear in the text that the numerical model used in this study is a finite-difference code solving non-linear shallow water equations in the Cartesian coordinate system. Please clarify . Also, I suppose that the numerical tsunami code (Goto et al., 1997) was benchmarked and used to accurately simulate other tsunami events, thereby, the authors should mention some references on this. . Which algorithm was used to track the shoreline movement and calculate the inundation? Is it the moving boundary algorithm (Liu et al., 1995)? Other? 4. METHOD FOR THE DEVELOPMENT OF EVACUATION PLANS: The paper addresses the development of tsunami evacuation plans using high-resolution flood maps and compares the estimated inundation depths with the buildings heights to define the vertical evacuation shelters. In my opinion, a crucial component for the development of effective evacuation plans is missing in this approach. It consists of investigating the vulnerability of the coastal building located

within the inundation zone, in particular, the buildings assessed as shelters. This must include an assessment of the buildings resistance capacity to a successive impact of both the earthquake and the tsunami. The study site is located within the co-seismic deformation area (Fig. 1 and 3) and, therefore, a Mw8.5-9.0 earthquake would cause a strong shaking that can have heavy damage on the coastal buildings and road network well before the arrival of the tsunami wave. This issue must be addressed for an effective planning of tsunami evacuation in Padang, Indonesia. 5. RESULTS: The results of tsunami hazard assessment (inundation maps) are of good quality and reflect, On the other hand, results on evacuation plans must be reassessed taking into account the comment #4. 6. DISCUSSION: The discussion must be reworked on the light of the new results and include the vulnerability of the shelters to a successive impact from the earthquake and then the tsunami.

---

## Referee Comment (RC2) · Anonymous Referee #2 · 26 Jul 2017

GENERAL COMMENTS The paper by Muhammad A. and coauthors reports an assessment of tsunami consequences on the coastal city of Padang (Indonesia), basing on stochastic simulations of tsunami sources, and evaluation of the consequences on the buildings that have been identified as vertical evacuation shelters. In this way, the authors assess the evacuation capability of the community in case of major tsunamis, the possible evacuation route and the time needed for people to evacuate, basing on the three selected earthquake-magnitude scenarios. An interesting issue is also represented by the evaluation of the effects of different levels of topographic data detail on the computation of tsunami inundation effects. The paper is in general very well written, well structured, and presents interesting results in the field of natural hazard

assessment and consequences on coastal communities. On the other side, some main issues need, on the referee opinion, to be address and clarified, as reported in the section below "Specific Comments", in order to be published on NHESS.

SPECIFIC COMMENTS 1. When dealing with building vertical evacuation, is it also considered the possibility of building collapses due to the earthquake itself? Such major earthquake often have considerable effects on edifice stability and integrity. 2. Explain the choice of the magnitudes (8.5-8.75-9) for the stochastic simulations. Does it mean that for lower values no tsunamis are generated? Provide some more details on tsunami numerical simulation (finite difference? Inundation with moving boundary?) 3. The probabilistic approach surely presents some advantages with respect to the deterministic one, taking into consideration also different possible features that the second cannot contemplate, but suffers from some main limitation: first of all, it can be applied only in coastal areas with a detailed knowledge of the seismic structures and a populated seismic and tsunami catalogue. Moreover, the paper refers to the 1797 event when reconstructing the fault geometry: for sure, it is one of the most reasonable mechanism, but it is not the only one and different events with different characteristics can produce different tsunamis. Please mitigate in general the sentences concerning the probabilistic vs deterministic approaches, highlighting also the problems of the first. The text repeatedly reminds that the deterministic approach produces over-simplification, but this is true for over-simplified applications of this methodology, not meaning that the whole procedure is wrong. 4. How do you expect authorities should use such probabilistic results? Can a decision-makers deal with scientific concepts like probability?

TECHNICAL CORRECTIONS Instead of using the word "depth" when referring the water column, use "flow depth". Line 43: Mueller et al paper year is 2015, not 2014 (ok in references) Line 78: "improve" instead of "improving" Line 160: "basing" instead of "based" Figures 9 to 12: what is intended for "inundation height in the coastal line"? Is it the height of the wave on the coast, before land flooding? Or is it the maximum

inland elevation reached by the water? In the first case it should be addressed as "maximum wave height on the coast", in the second it is simply "run-up height". Please clarify this point. Line 338: is the Padang population referred to an average value? Does this esteem take into account tourist period, seasonal variation and so on? Lines 372-3: "... to estimate the tsunami hazard level in Padang adopting three magnitude scenarios (Mw 8.5, Mw 8.75, and Mw 9.0)" FIGURES 3 to 8: use different palettes for the different figures, addressing different quantities (slip, land elevation, elevation difference, inundation-tsunami depth), it can create confusion.
* * *

---

## Author Comment (AC1) · 20 Sep 2017

We appreciate Reviewer 1 for his/her positive and constructive comments on the submitted manuscript. The following are our point-by-point responses to the reviewer's comments.

C1 - INTRODUCTION SECTION: In the introduction section, there is a lack of a description of the effects of the 26th December 2004 Indian Ocean Tsunami on the study area. Therefore, I suggest that the authors add a short paragraph in which they are invited to outline the reasons why the study area of Padang, Indonesia, escaped the destructive effects of the 26 December 2004 Indian Ocean tsunami.

Answer:

The main reason insignificant effects of the 2004 Indian Ocean tsunami in Padang areas is because the source location where the earthquake rupture occurred was far from Padang (i.e. >1,200 km). The 2004 source is centred in the Andaman segment of the Sunda subduction zone which is located in the north-west of Sumatra Islands (Meltzner et al., 2006; Briggs et al., 2006). On the other hand, Padang is located in the centre-west part of Sumatra and therefore, the 2004 tsunami events was not majorly affected this areas.

We added a short paragraph in Introduction describing the effects of the 2004 Indian Ocean tsunami to outline the 2004 Indian Ocean tsunami as follow:

The 2004 Indian Ocean tsunami did not significantly affect this region since the source location of the 2004 event is far i.e. >1,200 km (Natawidjaja et al., 2006; Meltzner et al., 2006; Briggs et al., 2006). However, it is located along the coast of Sumatra Island, directly facing the Mentawai segment of the Sunda subduction zone. Consequently, potential impact of the future tsunami may have significant risk in this area. In addition, with the low-lying plain topographic features in Padang, the probability of large inundated areas and large inundation depths is also high (Borrero et al., 2006; Muhari et al., 2010, 2011).

C2 - EARTHQUAKE SCENARIO SELECTION: The authors must justify their choice as regards the magnitudes of the earthquake scenarios. Why a minimum Magnitude of 8.5 and not 8.25, for instance? Is a Mw8.25 earthquake causes a tsunami with no significant effect on the study area? Why do the authors not consider a maximum earthquake magnitude in the range of this of the 2004 Indian Ocean event (Mw9.2)? Such a choice would change the predicted tsunami inundation characteristics and therefore the associated evacuation plan.

Answer:

We used Mw 8.5 as the minimum scenario magnitude in our study because the tsunami hazard produced from the magnitudes less than this level, e.g. Mw 8.25 and Mw 8.0, are considered to be relatively small (below 1 m wave height in the coastal areas; McCloskey et al., 2008; Muhammad et al., 2016). From Figure 1 in this document, we see the relatively minor effects in Padang due to Mw 8.5 tsunami . It shows the tsunami wave heights at three stations, i.e. Tabing, Purus, and Teluk Bayur, at a depth of 5 m. The median wave heights produced from 100 tsunamigenic scenarios are about 1 m which is small and will have minor impact on land (see Muhammad et al., 2016). The impact becomes more insignificant if we consider the Mw 8.25 scenario. Therefore, we choose the Mw 8.5 as the minimum magnitude scenarios.

For the maximum scenarios (i.e. Mw 9.0), it was selected based on the existing research studies from geodetic, paleogeodetic, and paleotsunami investigations. These studies indicated that the accumulated slip in the Mentawai segment of the Sunda subduction zone may generate the tsunamigenic earthquake with the magnitude ranging from Mw 8.8 to Mw 9.0 (Zachariasen et al., 1999; Natawidjaja et al., 2006; Sieh et al., 2008). We did not consider a extreme scenario like the 2004 Indian Ocean tsunami (which is very long) because the tsunami sediment records in North of Sumatra indicated that the recurrence time of destructive tsunamis from the Aceh-Andaman sources is at least 600 years in comparison to ~200 years for the Mentawai segment (Natawidjaja et al., 2006; Monecke et al., 2008) and hence, the Mw 9.0 consider to be more likely than the scenarios such as the 2004 Indian Ocean tsunami. However, such long ruptures are a possibility in the Mentawai segment – we simply did not consider such an assumption.

Based on Reviewer's recommendations, we have added to the methodology section of the revised manuscript the following descriptions regarding the choice of our magnitude scenarios in the revised manucript as follows:

The use of magnitude Mw 8.5 as the minimum scenario is because the tsunami hazard produced from the magnitude below this level, e.g. Mw 8.25 and Mw 8.0 is relatively

small (below 1 m wave height in the coastal areas; see Muhammad et al., 2016). The maximum magnitude scenario (Mw 9.0) is based on the recommendation from geodetic, paleogeodetic, and paleotsunami studies confirmed that the accumulated slip in the Mentawai segment of the Sunda subduction zone may generate the tsunamigenic earthquake with the magnitude range from Mw 8.8 to Mw 9.0 (Zachariasen et al., 1999; Natawidjaja et al., 2006; Sieh et al., 2008).

C3 - STOCHASTIC TSUNAMI SIMULATION: It is not clear in the text that the numerical model used in this study is a finite-difference code solving non-linear shallow water equations in the Cartesian coordinate system. Please clarify. Also, I suppose that the numerical tsunami code (Goto et al., 1997) was benchmarked and used to accurately simulate other tsunami events, thereby, the authors should mention some references on this. Which algorithm was used to track the shoreline movement and calculate the inundation? Is it the moving boundary algorithm (Liu et al., 1995)? Other?

Answer: Yes it is. We used a finite-difference code to numerically solve non-linear shallow water equations in the Cartesian coordinate system.

We have added several references to the revised manuscript regarding the implementation of Goto et al. (1997) model for tsunami simulation.

The algorithm to track the shoreline movement and calculate the inundation is using approximate moving boundary algorithm proposed by Iwasaki and Mano (1797).

To cover the recommendations from the reviewer, the following texts have been added into the revised manuscript:

A finite-difference method incorporating staggered leap-frog scheme is adopted to solve the governing equations (Goto et al., 1997). In addition, in Goto et al. (1997) code the moving boundary approach developed by Iwasako and Mano (1797) is used for inundation modelling. This method has been successfully used to run the tsunami simulation in several region including Padang, Indonesia, Mexico, and Japan (Muhari

et al., 2010, 2011; Goda et al., 2014; Mori et al., 2017).

C4 - METHOD FOR THE DEVELOPMENT OF EVACUATION PLANS: The paper addresses the development of tsunami evacuation plans using high-resolution flood maps and compares the estimated inundation depths with the buildings heights to define the vertical evacuation shelters. In my opinion, a crucial component for the development of effective evacuation plans is missing in this approach. It consists of investigating the vulnerability of the coastal building located within the inundation zone, in particular, the buildings assessed as shelters. This must include an assessment of the buildings resistance capacity to a successive impact of both the earthquake and the tsunami. The study site is located within the co-seismic deformation area (Fig. 1 and 3) and, therefore, a Mw8.5-9.0 earthquake would cause a strong shaking that can have heavy damage on the coastal buildings and road network well before the arrival of the tsunami wave. This issue must be addressed for an effective planning of tsunami evacuation in Padang, Indonesia.

Answer:

Thank you very much for these valuable comments. We have re-assessed the vulnerability of tsunami evacuation shelters (TES) considering both seismic and tsunami loadings. A new section: Section 2.2. vulnerability assessment of tsunami evacuation shelters has been added in the methodology section of the revised manuscript to explain the shaking and tsunami vulnerability assessments. The results from these assessments are also included in the results and discussion section. To facilitate the communications with the editor and the reviewers, a summary of the TES vulnerability assessment procedure and the assessments results is detailed in the supplement file (i.e. NHESS-2017-75-suplement.pdf). It is an extensive response so the PDF attachment is needed.

C5 - RESULTS: The results of tsunami hazard assessment (inundation maps) are of good quality and reflect, On the other hand, results on evacuation plans must be reassessed taking into account the comment #4.

C6 - DISCUSSION: The discussion must be reworked on the light of the new results and include the vulnerability of the shelters to a successive impact from the earthquake and then the tsunami. Answer C5 and C6:

We have incorporated the seismic and tsunami vulnerability assessment results to the results and discussion section of the revised manuscript.

Please also note the supplement to this comment:
https://www.nat-hazards-earth-syst-sci-discuss.net/nhess-2017-75/nhess-2017-75-AC1-supplement.pdf

———————————————————

[Figure]

**Fig. 1.** Tsunami wave height profile near coastal line of Padang: (A) site location. (B). Tabing (P1) station). (C). Purus (P2) station.

---

## Author Comment (AC2) · 20 Sep 2017

We highly appreciated the constructive comments given by the reviewer 2 for our submitted manuscript. The following are detail response of the reviewers comments. C1: When dealing with building vertical evacuation, is it also considered the possibility of building collapses due to the earthquake itself? Such major earthquake often have considerable effects on edifice stability and integrity.

Answer:

Agreed. Considering the reviewer recommendations, we have re-assessed the vul-

nerability of tsunami evacuation shelters (TES) considering both seismic and tsunami loadings. A new section Section 2.2. vulnerability assessment of tsunami evacuation shelters has been added in the methodology section of the revised manuscript to explain the shaking and tsunami vulnerability assessments. The results from these assessments are also included in the results and discussion section. To facilitate the communications with the editor and the reviewers, a summary of the TES vulnerability assessment procedure and the assessments results are detailed in the supplement file (i.e. NHESS-2017-75-suplement.pdf). It is an extensive response, so the PDF attachment is needed.

C2: Explain the choice of the magnitudes (8.5-8.75-9) for the stochastic simulations. Does it mean that for lower values no tsunamis are generated?

Answer:

We used Mw 8.5 as the minimum scenario magnitude in our study because the tsunami hazard produced from the magnitudes less than this level, e.g. Mw 8.25 and Mw 8.0, are considered to be relatively small (below 1 m wave height in the coastal areas; McCloskey et al., 2008; Muhammad et al., 2016). From Figure 1 in this document, we see the relatively minor effects in Padang due to Mw 8.5 tsunami . It shows the tsunami wave heights at three stations, i.e. Tabing, Purus, and Teluk Bayur, at a depth of 5 m. The median wave heights produced from 100 tsunamigenic scenarios are about 1 m which is small and will have minor impact on land (see Muhammad et al., 2016). The impact becomes more insignificant if we consider the Mw 8.25 scenario. Therefore, we choose the Mw 8.5 as the minimum magnitude scenarios.

For the maximum scenarios (i.e. Mw 9.0), it was selected based on the existing research studies from geodetic, paleogeodetic, and paleotsunami investigations. These studies indicated that the accumulated slip in the Mentawai segment of the Sunda subduction zone may generate the tsunamigenic earthquake with the magnitude ranging from Mw 8.8 to Mw 9.0 (Zachariasen et al., 1999; Natawidjaja et al., 2006; Sieh

et al., 2008). We did not consider a extreme scenario like the 2004 Indian Ocean tsunami (which is very long) because the tsunami sediment records in North of Suma-tra indicated that the recurrence time of destructive tsunamis from the Aceh-Andaman sources is at least 600 years in comparison to ∼200 years for the Mentawai segment (Natawidjaja et al., 2006; Monecke et al., 2008) and hence, the Mw 9.0 consider to be more likely than the scenarios such as the 2004 Indian Ocean tsunami. However, such long ruptures are a possibility in the Mentawai segment – we simply did not consider such an assumption.

Based on Reviewer's recommendations, we have added to the methodology section of the revised manuscript the following descriptions regarding the choice of our magnitude scenarios in the revised manucript as follows:

The use of magnitude Mw 8.5 as the minimum scenario is because the tsunami hazard produced from the magnitude below this level, e.g. Mw 8.25 and Mw 8.0 is relatively small (below 1 m wave height in the coastal areas; see Muhammad et al., 2016). The maximum magnitude scenario (Mw 9.0) is based on the recommendation from geode-tic, paleogeodetic, and paleotsunami studies confirmed that the accumulated slip in the Mentawai segment of the Sunda subduction zone may generate the tsunamigenic earthquake with the magnitude range from Mw 8.8 to Mw 9.0 (Zachariasen et al., 1999; Natawidjaja et al., 2006; Sieh et al., 2008).

C3: Provide some more details on tsunami numerical simulation (finite difference? Inundation with moving boundary?)

Answer:

Agreed. We have added the detail regarding numerical simulation in the revised manuscript which are the following:

A finite-difference method incorporating staggered leap-frog scheme is adopted to solve the governing equations (Goto et al., 1997). In addition, in Goto et al. (1997)

code the moving boundary approach developed by Iwasako and Mano (1797) is used for inundation modelling.

C4: The paper refers to the 1797 event when reconstructing the fault geometry: for sure, it is one of the most reasonable mechanism, but it is not the only one and different events with different characteristics can produce different tsunamis.

Answer: Agreed. The geodetic, paleogeodetic, and paleotsunami studies confirmed that two significant tsunamigenic events occurred in 1797 and 1833 events (Natawidjaja et al., 2006; Sieh et al., 2008; Philibosian et al., 2014) and hence, the scenario may not only follow the 1797 event. We absolutely aware that the possible event from the 1833 source may occur as well. Moreover, a significant tsunamigenic event generated from any point in the Sunda subduction zone is also possible. However, current literature has suggested that the tsunamigenic event from the 1797 scenario may produce the most devastating effects in Padang areas (Borrero et al., 2006; Natawidjaja et al., 2006; McCloskey et al., 2008; Muhari et al., 2010,2011; Griffin et al., 2016). The historical record regarding the effects of the 1833 and the 1797 events in Padang also confirmed that the 1797 produce more damage than the 1833 event (Natawidjaja et al., 2006). Subsequently, since we consider the worst scenario for the future event, the 1797 event is chosen.

We have a short description regarding the reason of choosing this scenarios in the revised manuscript: Note that the 1797 event was found to produce more significant tsunami impacts in Padang than the 1833 event (Borrero et al., 2006; Natawidjaja et al., 2006; McCloskey et al., 2008). Consequently, in this study, the 1797 asperity zone is adopted to generate the future stochastic earthquake source models.

C5: The probabilistic approach surely presents some advantages with respect to the deterministic one, taking into consideration also different possible features that the second cannot contemplate, but suffers from some main limitation: first of all, it can be applied only in coastal areas with a detailed knowledge of the seismic structures and

a populated seismic and tsunami catalogue. Please mitigate in general the sentences concerning the probabilistic vs deterministic approaches, highlighting also the problems of the first. The text repeatedly reminds that the deterministic approach produces oversimplification, but this is true for over-simplified applications of this methodology, not meaning that the whole procedure is wrong.

Agreed. We have added the following texts to outline this problem:

In the past, two types of earthquake source scenarios have been mainly considered to develop tsunami risk mitigation plans in Padang: deterministic scenarios (Borrero et al., 2006; Schlurmann et al., 2010; Muhari et al., 2010, 2011) and probabilistic scenarios (McCloskey et al., 2008; Griffin et al., 2016). These two methods have both advantages and disadvantages. For instance, the deterministic approach is more communicable to the authorities for developing post-hazard recovery and mitigation plans (McGuire, 2001). However, Implementation of deterministic scenarios may oversimplify the tsunami hazards and risks, leading to inaccurate mitigation plans (Mueller et al., 2014; Griffin et al., 2015). On the other hand, the probabilistic scenario approach requires the proper consideration of regional earthquake characteristics, including uncertainties in size of the rupture plane and spatial heterogeneity of earthquake slip. Therefore, extensive and detail data regarding the regional seismological characteristics are essential to adopt the probabilistic scenario.

Answer:

C6: How do you expect authorities should use such probabilistic results? Can a decision-makers deal with scientific concepts like probability?

Answer:

The work in this manuscript is a preliminary step to implement into more practical implementation for disaster risk reduction. The following work that may be carried out in the near future regarding our methodology is Probabilistic Tsunami Hazard Analysis (PTHA) in Padang, Indonesia considering the stochastic tsunami simulation. The PTHA may produce the tsunami hazard maps showing the annual probability of experiencing a tsunami with a specific tsunami intensity hazard, e.g. height, depth and velocity. Through this approach, we may effectively use to communicate with the authorities for improving the tsunami mitigation systems in Padang, Indonesia. Moreover, several preliminary works regarding the PTHA using the stochastic tsunami simulation have been successfully implemented in Japan and Mexico (De Risi et al., 2016 and Mori et al., 2017) and hence, it is possible to produce such results.

C7: Figures 9 to 12: what is intended for "inundation height in the coastal line"? Is it the height of the wave on the coast, before land flooding? Or is it the maximum inland elevation reached by the water? In the first case it should be addressed as "maximum wave height on the coast", in the second it is simply "run-up height". Please clarify this point.

Answer:

It is the maximum wave height on the coast. We have corrected in the revised manuscript: instead of only the inundation height along the coastal, we have changed to the maximum wave height on the coast.

C8: Line 338: is the Padang population referred to an average value? Does this esteem take into account tourist period, seasonal variation and so on?

Answer:

It is only the average value of Padang population without considering other condition, e.g. tourist period.

Subsequently, we have added these texts into the revised manuscript:

Noted that, the capacity (in persons) of the TES calculated in this study only consider the average population number of Padang excluding other conditions, e.g. tourist period and seasonal variation.

C9: TECHNICAL CORRECTIONS

(1) Instead of using the word "depth" when referring the water column, use "flow depth". Agreed. It has been changed accordingly. (2) Line 43: Mueller et al paper year is 2015, not 2014 (ok in references) (3) Line 78: "improve" instead of "improving" Line 160: "basing" instead of "based" (4) 372-3: ". . . to estimate the tsunami hazard level in Padang adopting three magnitude scenarios (Mw 8.5, Mw 8.75, and Mw 9.0) (5) " FIGURES 3 to 8: use different palettes for the different figures, addressing different quantities (slip, land elevation, elevation difference, inundation-tsunami depth), it can create confusion.

Answer: Agreed. The technical corrections have been included in the revised manuscripts.

Please also note the supplement to this comment:
https://www.nat-hazards-earth-syst-sci-discuss.net/nhess-2017-75/nhess-2017-75-AC2-supplement.pdf
* * *
[Figure]

**Fig. 1.** Tsunami wave height profile near coastal line of Padang: (A) site location. (B). Tabing (P1) station. (C). Purus (P2) station.

[Figure]

**Supplement:**

**PROCEDURE OF SHAKING AND TSUNAMI VULNERABILITY ASSESSMENTS OF TSUNAMI EVACUATION SHELTERS**

A procedure to carry out vulnerability assessments of TESs due to shaking and tsunami is presented in Figure 1. First, earthquake and tsunami simulations are conducted. A ground motion prediction equations (GMPE) developed by Abrahamson et al. (2016) is adopted to carry out the earthquake simulation. Noting that, the source scenarios for the seismic and tsunami simulations are the same. Second, the vulnerability assessment is carried out. The building vulnerability is assessed by determining the probability of a building experiencing specific damage states for a given hazard level, e.g. spectral acceleration (Sa) at a certain vibration period and spectral displacement for shaking and maximum tsunami depth (h) for tsunami (Rossetto and Elnashai, 2003; Sabandi et al., 2004; Porter et al., 2007; Ahmad et al., 2015; De Risi and Goda, 2016). Subsequently, the fragility models for both earthquake and tsunami vulnerability assessment are adopted. Since the tsunami is a secondary hazard triggered by an earthquake fault rupture, the seismic vulnerability assessment of TESs is carried out prior to the tsunami vulnerability assessment (see Figure 1). Detail procedures for the TES earthquake-tsunami hazard and vulnerability assessments are presented below.

[Figure]

Figure 1. Procedure of earthquake and tsunami hazard and vulnerability assessment of tsunami evacuation shelters.

**1. EARTHQUAKE SIMULATION**

Seismic intensity measures for a given earthquake scenario at the TEBs can be effectively estimated using ground motion prediction equations (Wald et al., 2016; Douglas, 2017). In this study, spectral acceleration (Sa) is selected as the intensity measure because most of empirical fragility models for buildings (e.g. HAZUS; Rossetto and Elnashai, 2003; Ahmad et al., 2014) use this parameter. Among existing GMPEs (e.g. Chang et al., 2001; Kataoka et al., 2006; Goda and Atkinson, 2009; Morikawa and Fujiwara, 2013; Abrahamson et al., 2016), a relationship by Abrahamson et al. (2016) is adopted for three reasons: (1) it is developed for global subduction environment (rather than specific geographical regions) and therefore, is applicable to mega-thrust interface subduction earthquakes in Sumatra, Indonesia, (2) including two of the most recent significant subduction earthquakes, i.e. the 2010 Maule, Chile and the 2011 Tohoku, Japan earthquakes, and (3) developed from an extensive ground motion dataset for subduction earthquakes around the world.

An equation to model the ground motion intensity of an interface subduction earthquake is the following:

$$\ln(S_a) = \theta_1 + \theta_4 \cdot \Delta C_1 + [\theta_2 + \theta_3 \cdot (M_w - 7.8)] \cdot \ln\{R + C_4 \cdot \exp[\theta_9 \cdot (M_w - 6)]\} + \theta_6 \cdot R +$$

$$f_{MAG}(M_w) + f_{FABA}(R) + f_{SITE}(PGA_{1000}, V_{s30}) + \sigma \cdot \varepsilon \tag{1}$$

where ln is the natural logarithm, Mw is … R (km) is the closest distance to the rupture area, $V_{S30}$ (m/s) is the average shear wave velocity in the uppermost 30 m of soil column, $PGA_{1000}$ is the mean peak ground acceleration (PGA) value corresponding to $V_{S30}$ = 1000 m/s, $\sigma$ is the total standard deviation (SD), and $\varepsilon$ is the Gaussian error term represented by zero mean and unit SD. The magnitude function is shown below:

$$f_{MAG}(M) = \begin{cases} \theta_4 \cdot [M_w - (7.8 + \Delta C_1)] + \theta_{13}.(10 - M)^2 & for\ M_w \leq 7.8 + \Delta C_1 \\ \theta_5 \cdot [M_w - (7.8 + \Delta C_1)] + \theta_{13}.(10 - M)^2 & for\ M_w \leq 7.8 + \Delta C_1 \end{cases} \tag{2}$$

where $\Delta C_1$ represents the epistemic uncertainty to control the magnitude scaling. $f_{FABA}(R)$ is the forearc/backarc terms which is equal to 0 for forearc or unknown site and one for backarc. Because Padang is in the forearc region of the Sumtara subduction zone, $f_{FABA}(R)$ is set to 0.

Finally, the site response scaling is presented by:

$$f_{SITE} = \theta_{12} \cdot \ln\left(\frac{\min(V_{S30}, 1000)}{V_{lin}}\right) - b \cdot \ln(PGA_{1000} + c) + b$$

$$\cdot \ln\left[PGA_{1000} + c \cdot \left(\frac{\min(V_{S30}, 1000)}{V_{lin}}\right)^n\right] \qquad for\ V_{S30} < V_{lin}$$

$$f_{SITE} = \theta_{12} \cdot \ln\left(\frac{\min(V_{S30}, 1000)}{V_{lin}}\right) + b \cdot n \cdot \ln\left(\frac{\min(V_{S30}, 1000)}{V_{lin}}\right) \qquad for\ V_{S30} \geq V_{lin} \qquad (3)$$

All model coefficients for Eqs 1-3 can be found in Abrahamson et al. (2016).

Three parameters are needed as input to simulate the interface megathrust earthquake including magnitude, rupture distance, and shear wave velocity for the considered site. For the TES assessment purposes, only the worst magnitude earthquake scenario is considered (Mw 9.0). On the other hand, the rupture distance is determined based on the closest distance between the location of interest and the rupture areas. Since the locations of TEBs are relatively close (the maximum distance among the TEB is less than 3 km) and this is significantly smaller than the distance between Padang and the rupture plane, seismic vulnerability assessment is conducted for a single representative site in Padang. Subsequently, the rupture distance to the site for the worst case is 55 km calculated from the smallest distance from the source to site of the 100 earthquake scenarios. On the other hand, the longest distance from the source to site among the 100 scenarios is ~100 km. Moreover, $V_{S30}$ in coastal areas of Padang ranges from 200 m/s to 400 m/s (Han et al., 2012; Putra et al., 2014) and hence, $V_{S30}$ of 300 m/s is used in this study. Finally, to include the uncertainty of the prediction equation for multiple spectral acceleration ordinates, a multivariate lognormal distribution is adopted. The median values of spectral acceleration at different vibration periods (at a site of interest) are evaluated using the GMPE with the three parameters, whereas their covariance are based on interperiod correlation of

ground motion parameters $\rho(T_1, T_2)$ (see Baker and Cornell, 2006). The correlation coefficient matrix has diagonal elements equal to 1 and off-diagonal elements equal to the correlation coefficient, $\rho$. It was calculated based on the following equation (Goda and Atkinson, 2009):

$$\rho(T_1, T_2) = \frac{1}{3}\left(1 - \cos\left\{\frac{\pi}{2} - \left[\theta_1 + \theta_2 I_{T_{min}<0.25} \times \left(\frac{T_{min}}{T_{max}}\right)^{\theta_3} \log_{10}\left(\frac{T_{min}}{0.25}\right)\right] \log_{10}\left(\frac{T_{max}}{T_{min}}\right)\right\}\right) + \frac{1}{3}\left\{1 + \cos\left[-1.5\log_{10}\left(\frac{T_{max}}{T_{min}}\right)\right]\right\}$$

$$(4)$$

where $\theta_1$, $\theta_2$, and $\theta_3$ are the model parameters ($\theta_1 = 1.374$, $\theta_2 = 5.586$, and $\theta_3 = 0.728$), $T_{max}$ and $T_{min}$ are the maximum and the minimum value of $T_1$ and $T_2$, respectively and $I_{T_{min}<0.25}$ is the indicator function that equals one if $T_{min} < 0.25$ sec and equals zero otherwise. Equation (4) was developed based on subduction earthquake records from Japan; thus, it is considered to be applicable to subduction earthquakes in Sumatra.

**2. SEISMIC AND TSUNAMI VULNERABILITY ASSESSMENTS**

For seismic vulnerability assessment, the fragility curves developed by Federal Emergency Management Agency (FEMA), i.e. HAZUS, is adopted to assess the vulnerability of TESs in Padang because of the following reasons:

1. New Indonesian Earthquake Resistance Building Code (SNI-1726: 2012) mostly adopted the U.S. seismic design documents, i.e. FEMA P7502009, regarding earthquake provisions for new building and other structures, and ASCE/SEI 7-10 for minimum design load criterion (SNI-1726: 2012; Wijayanti et al., 2015; Sengara et al., 2016; Douglas and Gkimprixis, 2017).

2. HAZUS is a well-established earthquake loss estimation framework and has been implemented in several earthquake prone-countries for seismic risk assessment purposes, e.g., Haiti, Puerto Rico, France, Romania, Austria and Indonesia (Kulmesh, 2010; Peterson and Small, 2012; Wijayanti et al., 2015; Sengara et al., 2016; Douglas and Gkimprixis, 2017).

[Figure]

Figure 2. Example of Capacity Spectrum Method used in HAZUS.

Figure 2 illustrates the procedure for developing an inelastic response (demand) spectrum from the elastic response (input) spectrum in HAZUS. First, the acceleration-period response spectrum is generated from the earthquake simulation. It is further converted into the Acceleration-Displacement Response Spectra (ADRS). This ADRS is then defined as the Elastic Response Spectrum (ERS). Second, the demand spectrum is calculated by dividing the ERS by the reduction factors (i.e. $R_A$ at periods of constant acceleration and $R_V$ at periods of constant velocity). Noting that the reduction factors in HAZUS are equal to the reciprocal of $SR_A$ and $SR_V$ in ATC-40 (Applied Technology Council, 1996). For essential and average buildings (type B) the $SR_A$ and $SR_V$ should be less than 2.27 and 1.79, respectively (ATC,1996). On the other hand, the TESs may be classified as type B based on the ATC-40 system and hence, the $R_A$ and $R_V$ should be less than 2.27 and 1.79, respectively. In this study, both $R_A$ and $R_V$ are set to 1.5 (Yeh et al., 2000; Lin and Chang, 2003; Casarotti et al., 2009; Monteiro et al., 2014).

Third, the capacity curve taken from HAZUS is overlaid to compare with the inelastic response spectrum (see blue line in Figure 2). The capacity curves in HAZUS are defined based on two

engineering parameters, e.g. yield and ultimate strengths characterising the nonlinear (pushover) behavior of 36 building types (e.g. wood frame building to steel moment resisting frame). The building type classifications in HAZUS are based on the building material (e.g. wood, concrete and steel) and height. Therefore, following the HAZUS classification, the TESs in Padang are reinforced concrete moment resistant frames (RC-MRF) with different building heights. TESs no. 13 and 16 are considered to be high-rise RC-MRF (C1H), whereas the rest of TESs are mid-rise RC-MRF (C1M). Moreover, four seismic design code classifications including Pre-Code, Low-Code, Moderate-Code, and High-Code are defined corresponding to the seismic zone. In terms of seismic design code classification, High-Code is considered to be applicable to TESs in Padang, because Padang is located in the high seismic zone corresponding to the High-Code in HAZUS and TESs have been designed and constructed to higher standards/quality than other normal buildings. In the following, the seismic vulnerability assessment of TESs is carried out by focusing upon C1M because the C1H type is typically stronger than the C1M (i.e. for the same shaking intensity, C1H buildings are expected to perform better than CM1).

Finally, fragility curves developed in HAZUS are used to define the damage functions of the building. The probability of being in or exceeding a given damage state is modelled by a cumulative lognormal distribution. Four damage states, i.e. slight, moderate, extensive, and complete, are defined in HAZUS (see Figure 3). Subsequently, to determine whether a TES can be used for post-earthquake tsunami evacuation purposes (not for shelters), the building is categorised into safe and unsafe by referring to existing tagging criteria (FEMA-356, 2000; HAZUS, 2003; Bazzurro et al., 2006) including (see Figure 3):

- Green tag: the building may have experienced onset damage but is safe for immediate occupancy. The none-to-slight damage state is applicable.
- Yellow tag: re-occupancy of the building is restricted and limited access only is allowed. Moderate-to-extensive damage state corresponds to this case.

- Red tag: the building is unsafe and no access is granted, and will be in complete or collapse damage state.

[Figure]

Figure 3. Fragility curves developed in HAZUS-MH (2001).

Based on the above tagging criteria, the tsunami evacuation building may be judged as unsafe for evacuation if the probability of extensive and complete damage states is over 50%. This assumption gives a 50-50 chance that the building may experience above or below extensive damage (Bazzurro et al., 2004, 2006). Moreover, the 50% probability of extensive or severer damage state is typically identified as the threshold value of a yellow tag in HAZUS that is adopted in this study (see Figure 3) and hence, may be regarded as the limit state to define the availability of evacuation buildings during the tsunami inundation.

For tsunami vulnerability assessment, Suppasri et al. (2011) models for RC structure is further used because of these reasons: 1) It was developed through extensive remote sensing and tsunami

survey data (i.e. ~5,000 points) in Banda Aceh and Thailand for the 2004 Indian Ocean tsunami; (2) the model is the most recent one among existing tsunami fragility models that are applicable to Sumatra, Indonesia; and (3) the model was successfully calibrated for the building vulnerability observed in the west coast of Thailand due to the 2004 Indian Ocean tsunami. These criteria are important because current situations of tsunami mitigation measures in Padang resemble those in Banda Aceh and Thailand more closely than situations in other regions. In Suppasri et al. (2011) model, only three damage states are adopted (see Figure 4) to generate fragility curves for reinforced concrete building including slight (DST1), moderate (DST2), and major/severe damage state (DST3). Using the calculated probability exceedance of each damage state, the TES is considered to be unsafe if the damage probability exceedance of major damage is above 50% (the major damage is assumed to be similar with the extensive damage in seismic damage state criteria).

[Figure]

Figure 4. Tsunami fragility models developed by Suppasri et al. (2011)

**3. RESULTS**

For seismic vulnerability assessment, first, the earthquake-HAZUS vulnerability assessment using the median response spectra of the worst cases (i.e. using the closest distance of the possible 300 earthquake scenarios with the $M_w$ 9.0) is presented in Figure 5. The median response spectra profile is

presented in Figure 5A. The ADRS is further calculated from the acceleration response spectrum as shown in Figure 5B. Using the ADRS, the capacity response spectrum method is implemented to determine the performance (demand) point (Figure 5C). The displacement performance point is estimated to be about 3 inches (7.6 cm) and then used to calculate the probability exceedance of damage states for a TES. Figure 5D shows that the sum of probabilities for extensive and complete damage states is ~7% and hence, the TES is considered to be safe for the median response spectra of the worst case.

[Figure]

Figure 5. The seismic-HAZUS TES vulnerability assessment using the worst scenario.

[Figure]

Figure 6. Earthquake simulation results from 100 tsunamigenic scenarios: (A). Spectral acceleration. (B). Peak Ground Acceleration (PGA).

[Figure]

Figure 7. Probability exceedance of extensive and complete damage states for 100 seismic events.

Second, the results from a 100 tsunamigenic earthquake scenarios are presented in Figure 6. The spectral acceleration profile shows how large the ground shaking occurred in Padang, Indonesia

due to the 100 tsunamigenic earthquakes from the Mentawai segment of the Sunda subduction zone. The spectral acceleration profile (Figure 6A) from the 100 earthquake events shows that the range of Sa in Padang is in between 0.3 g to 0.9 g for the period below 1 s. Moreover, the PGA values (Figure 6B) is the interval of 0.3 g to 0.9 g with the median of about 0.5 g. Using the simulated response spectra from those 100 earthquake scenarios, the TES vulnerability is further performed. Figure 7 presents the probability exceedance of extensive (blue dot) and complete (black dot) damage states and the sum of these two probabilities (red dot). A 50% probability line is drawn to see the threshold of safe building. Figure 8 confirms that the TES may be operational for evacuation because ~95% from the total of 100 earthquake simulations produce less than 50% probability exceedance of above extensive damage state. Moreover, most of the cases results in less than 25% probability of exceedance above extensive damage state. Subsequently, the TES may be considered to be safe for evacuation after the ground shaking and hence, the tsunami vulnerability assessment can be carried out.

Fourth, the tsunami vulnerability assessment is performed. Using the maximum inundation depths at all 23 TESs from the 100 earthquake scenarios of the $M_w$ 9.0, the probability of exceeding severe damage state (DST3) of each TES is calculated. In this study, the number of event producing the probability of exceeding severe damage state of more than 50% is defined as destructive event. The destructive events for the shelter numbers 16 and 17 are relatively large, i.e. 30% and 36% of the 100 events are destructive for the shelter numbers 16 and 17, respectively and hence, these two shelters may be considered to be unsafe for the evacuation. Moreover, the percentage of non-destructive event for the other shelters are above 75%. Therefore, except for shelter number 16 and 17, the rest of the shelters consider to be operational for evacuation.

**References**

Abrahamson, N., Gregor, N., and Addo, K.: BC hydro ground motion prediction equations for subduction earthquakes. Earthquake Spectra 32, 23–44. doi: 10.1193/051712EQS188MR, 2016.

Applied Technology Council, : Seismic Evaluation and Retrofit of Concrete Buildings, Report ATC 40, November, 1996.

Ahmad, S., Kyriakides, N., Pilakoutas, K., Neocleous, K., and Zaman, Q. : Seismic fragility assessment of existing sub-standard low strength reinforced concrete structures, Earthq. Eng. Eng. Vib., 14, 439. https://doi.org/10.1007/s11803-015-0035-0, 2015.

Baker, J. W. and Cornell, C. A.: Correlation of response spectral values for multicomponent ground motions, Bull. Seismol. Soc. Am., 96, 215-227.

Bazzurro, P, Cornell, C. A., Menun, C, and Motahari, M.: Guidelines for seismic assessment of damaged buildings, the 13th World Conference on Earthquake Engineering, Vancouver, Canada, 1-6 August, 2004.

Bazzurro, P, Cornell, C. A., Menun, C, and Motahari, M.: Advanced seismic assessment guidelines, Pacific Earthquake Engineering Research Centre, Stanford University, 2006.

Briggs, R.W., Sieh, K., Meltzner, A.J., Natawidjaja, D., Galetzka, J., Suwargadi, B., Hsu, Y., Simons, M., Hananto, N., Suprihanto, I., Prayudi, D., Avouac, J.P., Prawirodirdjo, L., and Bock, Y. (2006). Deformation and slip along the Sunda megathrust in the great 2005 Nias-Simeulue earthquake. Science 311, 1897–1901. doi:10.1126/science.1122602.

Borrero, J. C., Sieh, K., Chlieh, M., and Synolakis, C. E.: Tsunami inundation modeling for western Sumatra, Proc. Nat. Acad. Sci. U.S.A., 103, 19673–19677, doi:10.1073/pnas.0604069103, 2006.

De Risi, R. and Goda, K.: Probabilistic Earthquake–Tsunami Multi-Hazard Analysis: Application to the Tohoku Region, Japan. Front. Built Environ., 2, 25, doi: 10.3389/fbuil.2016.00025, 2016.

Douglas, J and Gkimprixis, A.: Using targeted risk in seismic design codes: a summary of the state of the art and outstanding issues, 6th National Conference on Earthquake Engineering and 2nd National Conference on Earthquake Engineering and Seismology, Bucharest, Romania, 14/06/17 - 17/06/17, 3-10, 2017.

FEMA (Federal Emergency Management Agency) 356: Pre-standard and commentary for the seismic rehabilitation of buildings, Federal Emergency Management Agency, Washington, D.C., 2009.

HAZUS. Multi-hazard Loss Estimation Methodology Earthquake Model. Federal Emergency Management Agency, Washington, D.C., 2003.

Iwasaki, T. and Mano, A.: Two-dimensional numerical computation of tsunami run ups in Eulerian description, Proc. Of 26th Conference on Coastal Engineering, Japan, JSCE, 70-74.

Meltzner, A.J., Sieh, K., Abrams, M., Agnew, D. C., Hudnut, K.W., Avouac, J.P., and Natawidjaja, D. H. (2006). Uplift and subsidence associated with the great Aceh-Andaman earthquake of 2004. J. Geophys. Res. 111, B02407, doi:10.1029/2005JB003891.

Monecke, K., Finger, W., Klarer, D., Kongko, W., McAdoo, B. G., Moore, A. L., and Sudrajat, S.U. (2008). A 1,000-year sediment record of tsunami recurrence in northern Sumatra. Nature 455, 1232–1234. doi:10.1038/nature07374.

Mori N, Muhammad A, Goda K, Yasuda T and Ruiz-Angulo A (2017) Probabilistic Tsunami Hazard Analysis of the Pacific Coast of Mexico: Case Study Based on the 1995 Colima Earthquake Tsunami. Front. Built Environ. 3:34. doi: 10.3389/fbuil.2017.00034.

Muhari, A., Imamura, F., Koshimura, S., and Post, J.: Examination of three practical run-up models for assessing tsunami impact on highly populated areas, Nat. Hazards Earth Syst. Sci., 11, 3107-3123, doi:10.5194/nhess-11-3107-2011, 2011.

Muhari, A., Imamura, F., Natawidjaja, D. H., Diposaptono, S., Latief, H., Post, J., and Ismail, F.A.: Tsunami mitigation efforts with pTA in West Sumatra Province, Indonesia. J. Earthq. Tsunami, 4, 341–368, doi:10.1142/S1793431110000790, 2010.

Natawidjaja, D.H., Sieh, K., Chlieh, M., Galetzka, J., Suwargadi, B.W., Cheng, H., Edwards, R.L., Avouac, J.P., and Ward, S.N. (2006). Source parameters of the great Sumatran megathrust earthquakes of 1797 and 1833 inferred from coral microatolls. J. Geophys. Res. 111, B06403, doi:10.1029/2005JB004025.

Rossetto, T. and Elnashai, A.: Derivation of vulnerability functions for European-type RC structures based on observational data, Eng. Str., 25, 1241-1263, 2003.

Sarabandi, P., Pachakis, D., King, S., and Kiremidjian, A.: Empirical fragility functions from recent earthquakes. The 13th world conference in Earthquake Engineering, Vancouver, Canada, August 1-6, 2004.

Sengara, I. W, Sidi, I. D, Mulia, A, Muhammad, A, and Daniel H.: Development of risk coefficient for input to new Indonesian seismic building codes. Journal of Engineering and Technological Sciences, 48, 49–65, 2016.

Sieh, K., Natawidjaja, D.H., Meltzner, A.J., Shen, C.C., Cheng, H., Li, K.S., Suwargadi, B.W., Galetzka, J., Philibosian, B., and Edwards, R.L. (2008). Earthquake super cycles inferred from sea-level changes recorded in the corals of West Sumatra. Science 322, 1674–1678. doi:10.1126/science.1163589.

Wijayanti, E, Kristiawan, S. A, Purwanto, E, and Sangadji, S.: Seismic Vulnerability of Reinforced Concrete Building Based on the Development of Fragility Curve: A Case study, Applied Mechanics and Materials, 845, 252-258, 2015.

Zachariasen, J., Sieh, K., Taylor, F.W., Edwards, R.L., and Hantoro, W.S. (1999). Submergence and uplift associated with the giant 1833 Sumatran subduction earthquake: Evidence from coral microatolls. J. Geophys. Res. 104, 895–919.

---

## Author Response (AR1)

**REVIEWER 1**

We appreciate Reviewer 1 for his/her positive and constructive comments on the submitted manuscript. The following are our point-by-point responses to the reviewer's comments.

C1 - INTRODUCTION SECTION: In the introduction section, there is a lack of a description of the effects of the 26th December 2004 Indian Ocean Tsunami on the study area. Therefore, I suggest that the authors add a short paragraph in which they are invited to outline the reasons why the study area of Padang, Indonesia, escaped the destructive effects of the 26 December 2004 Indian Ocean tsunami.

**Answer:**

The main reason insignificant effects of the 2004 Indian Ocean tsunami in Padang areas is because the source location where the earthquake rupture occurred was far from Padang (i.e. >1,200 km). The 2004 source is centred in the Andaman segment of the Sunda subduction zone which is located in the north-west of Sumatra Islands (Meltzner et al., 2006; Briggs et al., 2006). On the other hand, Padang is located in the centre-west part of Sumatra and therefore, the 2004 tsunami events was not majorly affected this areas.

We added a short paragraph in Introduction describing the effects of the 2004 Indian Ocean tsunami to outline the 2004 Indian Ocean tsunami as follow:

The 2004 Aceh-Andaman tsunami did not significantly affect this region since the source location of the 2004 event was far, i.e. >1,200 km (Natawidjaja et al., 2006; Meltzner et al., 2006; Briggs et al., 2006). However, it is located along the coast of Sumatra Island, directly facing the Mentawai segment of the Sunda subduction zone. Consequently, potential impact of the future tsunami may be significant in this area. In addition, with the low-lying plain topographic features in Padang, the probability of large inundated areas and large inundation depths is also high (Borrero et al., 2006; Muhari et al., 2010, 2011).

C2 - EARTHQUAKE SCENARIO SELECTION: The authors must justify their choice as regards the magnitudes of the earthquake scenarios. Why a minimum Magnitude of 8.5 and not 8.25, for instance? Is a Mw8.25 earthquake causes a tsunami with no significant effect on the study area? Why do the authors not consider a maximum earthquake magnitude in the range of this of the 2004 Indian Ocean event (Mw9.2)? Such a choice would change the predicted tsunami inundation characteristics and therefore the associated evacuation plan.

**Answer:**

We used  $M_w$  8.5 as the minimum scenario magnitude in our study because the tsunami hazard produced from the magnitudes less than this level, e.g.  $M_w$  8.25 and  $M_w$  8.0, are considered to be relatively small (below 1 m wave height in the coastal areas; McCloskey et al., 2008; Muhammad et al., 2016). From Figure 1 in this document, we see the relatively minor effects in Padang due to  $M_w$  8.5 tsunami . It shows the tsunami wave heights at three stations, i.e. Tabing, Purus, and Teluk Bayur, at a depth of 5 m. The median wave heights produced from 100 tsunamigenic scenarios are about 1 m which is small and will have minor impact on land (see Muhammad et al., 2016). The impact becomes more insignificant if we consider the  $M_w$ 8.25 scenario. Therefore, we choose the  $M_w$  8.5 as the minimum magnitude scenarios.

Figure 1. Tsunami wave height profile near coastal line of Padang: (A) site location. (B). Tabing (P1) station). (C). Purus (P2) station.

For the maximum scenarios (i.e.  $M_w$  9.0), it was selected based on the existing research studies from geodetic, paleogeodetic, and paleotsunami investigations. These studies indicated that the accumulated slip in the Mentawai segment of the Sunda subduction zone may generate the tsunamigenic earthquake with the magnitude ranging from  $M_w$  8.8 to  $M_w$  9.0 (Zachariasen et al., 1999; Natawidjaja et al., 2006; Sieh et al., 2008). We did not consider a extreme scenario like the 2004 Indian Ocean tsunami (which is very long) because the tsunami sediment records in North of Sumatra indicated that the recurrence time of destructive tsunamis from the Aceh-Andaman sources is at least 600 years in comparison to ~200 years for the Mentawai segment (Natawidjaja et al., 2006; Monecke et al., 2008) and hence, the  $M_w$  9.0 consider to be more likely than the scenarios such as the 2004 Indian Ocean tsunami. However, such long ruptures are a possibility in the Mentawai segment – we simply did not consider such an assumption.

Based on Reviewer's recommendations, we have added to the methodology section of the revised manuscript the following descriptions regarding the choice of our magnitude scenarios in the revised manucript as follows:

The magnitude  $M_w$  8.5 is used as the minimum scenario because the tsunami hazard produced from the magnitude below this level, e.g.  $M_w$  8.25 and  $M_w$  8.0, is relatively small (less than 1 m wave height in the coastal areas; see Muhammad et al., 2016). The maximum magnitude scenario ( $M_w$  9.0) is selected based on geodetic, paleo-geodetic, and paleo-tsunami studies (Zachariasen et al., 1999; Natawidjaja et al., 2006; Sieh et al., 2008); they indicated that the accumulated slip in the Mentawai segment of the Sunda subduction zone may generate the tsunamigenic earthquake with the magnitude range from  $M_w$  8.8 to  $M_w$  9.0.

C3 - STOCHASTIC TSUNAMI SIMULATION: It is not clear in the text that the numerical model used in this study is a finite-difference code solving non-linear shallow water equations in the Cartesian coordinate system. Please clarify. Also, I suppose that the numerical tsunami code (Goto et al., 1997) was benchmarked and used to accurately simulate other tsunami events, thereby, the authors should mention some references on this. Which algorithm was used to track the shoreline movement and calculate the inundation? Is it the moving boundary algorithm (Liu et al., 1995)? Other?

**Answer:**

Yes it is. We used a finite-difference code to numerically solve non-linear shallow water equations in the Cartesian coordinate system.

We have added several references to the revised manuscript regarding the implementation of Goto et al. (1997) model for tsunami simulation.

The algorithm to track the shoreline movement and calculate the inundation is using approximate moving boundary algorithm proposed by Iwasaki and Mano (1797).

To cover the recommendations from the reviewer, the following texts have been added into the revised manuscript:

A finite-difference method implementing a staggered leap-frog scheme is adopted to solve the governing equations (Goto et al., 1997). In addition, in Goto et al.'s code the moving boundary approach developed by Iwasaki and Mano (1979) is used for inundation modeling. This method has been successfully used to run the tsunami simulation in several regions, including Padang, Indonesia, Mexico, and Japan (Muhari et al., 2010, 2011; Goda et al., 2014; Mori et al., 2017).

C4 - METHOD FOR THE DEVELOPMENT OF EVACUATION PLANS: The paper addresses the development of tsunami evacuation plans using high-resolution flood maps and compares the estimated inundation depths with the buildings heights to define the vertical evacuation shelters. In my opinion, a crucial component for the development of effective evacuation plans is missing in this approach. It consists of investigating the vulnerability of the coastal building located within the inundation zone, in particular, the buildings assessed as shelters. This must include an assessment of the buildings resistance capacity to a successive impact of both the earthquake and the tsunami. The study site is located within the co-seismic deformation area (Fig. 1 and 3) and, therefore, a Mw8.5-9.0 earthquake would cause a strong shaking that can have heavy damage on the coastal buildings and road network well before the arrival of the tsunami wave. This issue must be addressed for an effective planning of tsunami evacuation in Padang, Indonesia.

**Answer:**

Thank you very much for these valuable comments. We have re-assessed the vulnerability of tsunami evacuation shelters (TES) considering both seismic and tsunami loadings. A new section **Section 2.2. vulnerability assessment of tsunami evacuation shelters** has been added in the methodology section of the revised manuscript to explain the shaking and tsunami vulnerability assessments. The results from these assessments are also included in the results and discussion section. To facilitate the communications with the editor and the reviewers, a summary of the TES vulnerability assessment procedure and the assessments results is detailed in the following.

**PROCEDURE OF SHAKING AND TSUNAMI VULNERABILITY ASSESSMENTS OF TSUNAMI EVACUATION SHELTERS**

A procedure to carry out vulnerability assessments of TESs due to shaking and tsunami is presented in Figure 2. First, earthquake and tsunami simulations are conducted. A ground motion prediction equations (GMPE) developed by Abrahamson et al. (2016) is adopted to carry out the earthquake simulation (see in the revised manuscript section 2.2.1). Noted that,

the source scenarios for the seismic and tsunami simulations are the same. Second, the vulnerability assessment is carried out. The building vulnerability is assessed by determining the probability of a building experiencing specific damage states for a given hazard level, e.g. spectral acceleration (Sa) at a certain vibration period and spectral displacement for shaking and maximum tsunami depth (h) for tsunami (Rossetto and Elnashai, 2003; De Risi and Goda, 2016). Subsequently, the fragility models for both earthquake and tsunami vulnerability assessment are adopted. Since the tsunami is a secondary hazard triggered by an earthquake fault rupture, the seismic vulnerability assessment of TESs is carried out prior to the tsunami vulnerability assessment (see Figure 2). In this study, the combined effects of earthquake shaking and tsunami are not taken into account, because such multi-hazard fragility models are not available for TES in Padang. Detail procedures for the TES earthquake-tsunami hazard and vulnerability assessments are presented below.